# Pax3 cooperates with Ldb1 to direct local chromosome architecture during myogenic lineage specification

Alessandro Magli [1,2], June Baik[1,5], Pruthvi Pota[1,5], Carolina Ortiz Cordero[1], Il-Youp Kwak [1], Daniel J. Garry[1], Paul E. Love[3], Brian D. Dynlacht[4] & Rita C.R. Perlingeiro[1,2]

Chromatin looping allows enhancer-bound regulatory factors to influence transcription. Large domains, referred to as topologically associated domains, participate in genome organization. However, the mechanisms underlining interactions within these domains, which control gene expression, are not fully understood. Here we report that activation of embryonic myogenesis is associated with establishment of long-range chromatin interactions centered on Pax3-bound loci. Using mass spectrometry and genomic studies, we identify the ubiquitously expressed LIM-domain binding protein 1 (Ldb1) as the mediator of looping interactions at a subset of Pax3 binding sites. Ldb1 is recruited to Pax3-bound elements independently of CTCF-Cohesin, and is necessary for efficient deposition of H3K4me1 at these sites and chromatin looping. When Ldb1 is deleted in Pax3-expressing cells in vivo, specification of migratory myogenic progenitors is severely impaired. These results highlight Ldb1 requirement for Pax3 myogenic activity and demonstrate how transcription factors can promote formation of sub-topologically associated domain interactions involved in lineage specification.

[1] Department of Medicine, Lillehei Heart Institute, University of Minnesota, Minneapolis, MN 55455, USA. [2] Stem Cell Institute, University of Minnesota, Minneapolis, MN 55455, USA. [3] Section on Hematopoiesis and Lymphocyte Biology, Eunice Kennedy Shriver National Institute of Child Health and Human Development, National Institutes of Health, Bethesda, MD 20892, USA. [4] Department of Pathology, New York University Cancer Institute, New York University School of Medicine, New York, NY 10016, USA. [5] These authors contributed equally: June Baik, Pruthvi Pota. Correspondence and requests for materials should be addressed to A.M. (email: amagli@umn.edu) or to R.C.R.P. (email: perli032@umn.edu)

Chromatin topology is an essential element of gene regulation because it establishes the framework for nuclear chromosome organization and ultimately, interactions between distal regulatory elements (enhancers) and gene promoters (reviewed by Denker and de Laat[1]). In the last decade, unbiased genome-wide analysis of chromosome conformation using Hi-C has identified that different parts of the genome with similar histone marks form two compartments characterized either by active transcription, being gene rich and having higher chromatin accessibility (type A), or being gene poor, displaying lower gene expression and repressive histone marks (type B)[2]. Importantly, these studies demonstrated the existence of topologically associated domains (referred also as contact domains), which are chromosomal subunits displaying a higher contact frequency and defined by the binding of the Ctcf-Cohesin complex at the contact domain border[3–5].

Recently, three independent laboratories demonstrated that loss of CTCF-Cohesin disrupts contact domains and causes loss of interaction peaks within these structures[6–8]. Despite the elimination of contact domains, strengthening of the compartmentalization and relatively mild changes in gene expression were observed, suggesting that in their absence, chromatin status may mediate the interaction between different loci with similar marks which preserves the gene expression program. Using polymer simulations, the interplay between compartments and contact domain formation demonstrated these two structures compete with each other, thus supporting the model of Cohesin-dependent loop extrusion[9]. Moreover, high-resolution analysis of the *Drosophila* genome identified long-range interactions between loci with similar epigenetic marks[10], and demonstrated that the transcriptional state represents a major predictor of chromatin organization[11]. Combined with the observation that contact domains are highly conserved among multiple cell types[3,12], these data suggest that histone posttranslational modifications and enhancer–promoter interactions at a sub-contact domain scale may represent the main drivers responsible for the activation of specific gene expression programs.

Despite the existence of loci where looping interactions control gene expression (e.g., LCR:β-globin and the Bithorax locus[13,14]), the extent to which transcription factors (TF) shape the three-dimensional organization of the genome during differentiation is not clearly defined. In fact, while the ubiquitously expressed Yin Yang 1 (YY1) has been shown to mediate distinct enhancer–promoter interactions independently of CTCF in multiple cell types[15], only a few studies have investigated the mechanisms underlying the establishment of tissue-specific looping using a model of lineage specification. In situ Hi-C during macrophage activation identified a correlation between AP1 occupancy and establishment of new looping interactions[16]. Similarly, B cell activation requires Myc for the shift from long- to short-range interactions, which in turn facilitate enhancer–promoter contacts regulating gene expression[17]. More recently, Monahan and colleagues reported that increased expression of the olfactory receptor genes observed during olfactory neuron differentiation involves strengthening of intra- and inter-chromosomal interactions between the selected gene promoter and several enhancers bound by the Lhx2-Ebf-Ldb1 complex[18].

To systematically dissect TF-mediated regulation of looping, here we use the skeletal myogenic lineage as a model to study tissue-specific chromatin architecture induced by the transcription factor Pax3. Using a combination of differentiating cultures of doxycycline-inducible mouse embryonic stem (mES) cells and next-generation sequencing-based technologies, we find that Pax3-mediated activation of the myogenic program occurs through a time-dependent establishment of long-range interactions involving PAX3 binding sites. PAX3 genomic occupancy is associated with an increased deposition of histone marks (H3K4me1 and H3K27Ac) normally found at active enhancer regions, and overlaps to elements capable of driving gene expression in developing embryos. Using mass spectrometry, we then identify PAX3 interaction with members of the chromatin looping complex, including the LIM-domain binding protein 1 (LDB1). We demonstrate that LDB1 is recruited to a subset of PAX3-bound elements characterized by increased levels of H3K4me1 deposition. Reduced Ldb1 expression impairs Pax3-dependent myogenic specification both in vitro and in vivo, and decreases deposition of H3K4me1 and chromatin looping of PAX3-bound enhancers. Importantly, our study show that forced recruitment of LDB1 to PAX3 enhancers is sufficient to induce gene expression, chromatin looping and H3K4me1 deposition, thus supporting that changes in genomic architecture are capable of driving transcription of Pax3 target genes during myogenesis.

## Results

**Pax3-bound elements establish long-range interactions.** Doxycycline-controlled Pax3 expression in differentiating mouse embryonic stem cells enables the robust activation of the skeletal myogenic program[19] (Fig. 1a and Supplementary Fig. 1a–d). To understand the functional mechanism of Pax3 in this process, we performed Chromatin-immunoprecipitation followed by sequencing (ChIP-seq), using an anti-PAX3 antibody, in mesodermal cells (1-day induction) and myogenic progenitors (6-days induction)[20]. Globally, this approach revealed 3780 and 5710 PAX3 peaks in mesodermal cells and myogenic progenitors, respectively. Among these, we identified known PAX3 binding sites, such as the −111 kb and −57 kb elements controlling *Myf5* expression, a well-known Pax3 target gene during embryonic myogenesis[21,22] (Fig. 1b and Supplementary Fig. 1e, f). As observed with other transcription factors[23], comparison of PAX3 genomic occupancy using different datasets (myoblasts and fibroblasts) showed common and cell-type specific PAX3-bound loci[20]. Upon annotation of both datasets, we observed that the vast majority (~90%) of PAX3 ChIP-seq peaks are more than 5 kilobases (>5 kb) away from the transcription start site (TSS) of the nearest annotated gene (Fig. 1c). Based on these data, we speculated that Pax3 myogenic activity relies on the establishment of interactions (hereafter referred as long-range chromatin interactions or looping interactions) between distal PAX3 sites, such as the −111 kb enhancer controlling Myf5 expression, and other elements occupied by PAX3 or sites of transcription initiation (TSS).

To address this open question, we performed HiChIP[24], which combines the intra-nuclear proximity ligation of the Hi-C with the ChIP-seq protocol, in Pax3-induced mesodermal cells (1-day) and myogenic progenitors (6-days), along with control non-induced cells. After subtracting the HiChIP contacts called in non-induced Pax3 cells, we identified 3227 and 7987 loops in Pax3-induced mesodermal cells and myogenic progenitors, respectively (Fig. 1d and Supplementary Data 1). Using the genomic coordinates defining beginning and end of each loop (referred hereafter as loop anchors), we observed an overlap of 439/3780 (12%) and 5153/5710 (90%) between 1-day and 6-day PAX3 binding sites, respectively (Fig. 1d). Functional annotation of the loop anchors demonstrated enrichment for multiple biological process categories associated with embryonic and muscle development (Supplementary Fig. 1g, h). Furthermore, 140/442 (33%) and 1615/4647 (35%) of 1-day and 6-day differentially expressed genes[20], respectively, overlap with anchor-annotated genes (Supplementary Fig. 1i, j and Supplementary Data 2). Accordingly, visual analysis of the HiChIP

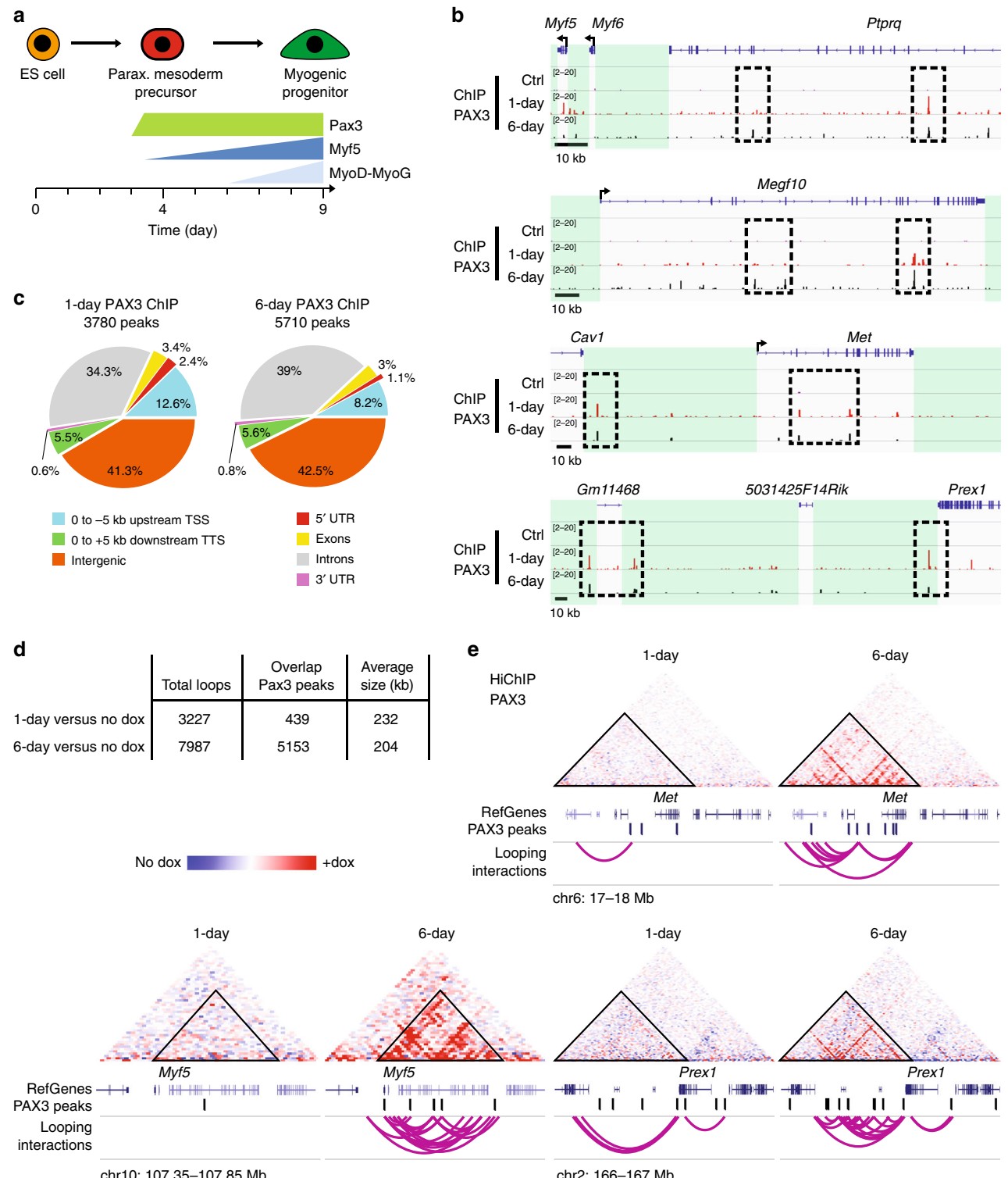

**Fig. 1** Pax3-bound elements establish long-range interactions. **a** Schematic representation of Pax3-induced myogenesis during mouse ES cell differentiation. Pax3 is induced at day 3 of differentiation using a doxycycline-dependent system. Twenty-four hours post Pax3-induction (1-day), only Myf5 transcript is detectable. Myod and Myog are expressed in the later stages of myogenic commitment (transcripts are detectable following 3/4-day Pax3 induction). **b** IGV track displaying PAX3 binding at *Myf5, Megf10, Met*, and *Prex1* loci in 1-day and 6-day Pax3-induced cells. Dashed black squares indicate PAX3-bound loci. Green boxes represent intergenic regions. Black arrow represents the transcription start site. **c** Functional annotation of PAX3 peaks from 1-day and 6-day Pax3-induced cells. **d** Number of long-range interactions (loops) detected by PAX3 HiChIP. **e** Long-range interactions at the *Met, Myf5*, and *Prex1* loci in 1-day and 6-day Pax3-induced cultures. The contact maps were normalized for sequencing depth and visualized as ratio relative to the background (non-induced PAX3 HiChIP). Triangles indicate domains of long-range chromatin interactions involving PAX3-bound sites (position of PAX3 peaks, genes, and chromosome coordinates are showed below). Scale: maxrange = 0.2. Arcs indicate looping interactions identified by FitHiChIP

matrices identified chromatin looping between PAX3-bound elements and distal chromatin loci (Fig. 1e). We observed that the transition from mesoderm to myogenic progenitors is associated with the appearance of clearly defined long-range interactions, as showed for the *Met* and *Prex1* loci (Fig. 1e). Chromatin looping was also observed for several other genes, including *Boc*, *Cdon*, *Dbx1*, *Megf10*, *Myf5*, and *Prrx1* (Fig. 1e and Supplementary Fig. 2a and see below). Establishment of long-range interactions at these sites coincided with a time-dependent increase in gene expression (Supplementary Fig. 2b), and in the case of the *Myf5* locus, epigenetic remodeling at the TSS region (*Myf5* +0.7 kb— Supplementary Fig. 2c). Thus, these results demonstrate that Pax3-mediated activation of the myogenic program is paralleled by the establishment of chromatin looping involving PAX3 bound distal elements.

**Epigenetic reprogramming of PAX3 binding sites**. To investigate the role of epigenetic marks in Pax3-mediated myogenic specification, we performed ChIP-seq using antibodies for H3K27me3, H3K27Ac, H3K4me1, and H3K4me3 in Pax3-induced mesodermal cells and myogenic progenitors. These posttranslational modifications of the histone 3 tail are widely used to identify repressed elements (H3K27me3), poised (H3K27me3 and H3K4me3), and active (H3K27Ac and H3K4me3) promoters and enhancers (H3K27Ac and H3K4me1)[25]. Visual inspection of the *Myf5* region and other genomic loci characterized by long-range chromatin interactions upon Pax3 induction showed an increased deposition of the epigenetic marks associated with gene activation, in agreement with our ChIP-qPCR data for H3K4me3 (Fig. 2a and Supplementary Fig. 2b, c). To determine whether Pax3 regulates the epigenetic status of its target loci, we compared the profile of H3K4me1, H3K4me3, H3K27Ac, and H3K27me3 deposition at PAX3 binding sites between non-induced and 1-day Pax3-induced mesodermal cells. Using a k-means clustering algorithm, PAX3 binding sites can be categorized into four distinct types of chromatin state (clusters 1–4 in Fig. 2b and Supplementary Fig. 3a), which displayed also different genomic distribution relative to the closest annotated gene (Supplementary Fig. 3b and Supplementary Data 3). PAX3 had no effect on the epigenetic status of clusters 2 and 3 loci (565 and 135 peaks, respectively—Supplementary Fig. 3c), which based on their histone marks profile can be categorized as elements proximal to TSS and repressed loci. In contrast, Pax3 induction resulted in increased deposition of H3K4me1 and a modest increase in H3K27Ac at clusters 1 and 4 loci (959 and 2121 peaks, respectively—Fig. 2c). PAX3 binding at cluster 1 elements was associated with increased bimodal distribution of H3K4me1 and H3K27Ac (Fig. 2c), a phenomenon observed at active enhancers[26]. On the other hand, cluster 4 sites displayed a clear increase in the levels of H3K4me1 but low H3K27Ac, thus suggesting this group may represent enhancers undergoing Pax3-mediated activation (Fig. 2c). Clusters 1 and 4 also showed a small but detectable increase in H3K4me3 levels (Fig. 2c). These results were consistent across biological replicates (Supplementary Fig. 4a). In agreement with Pax3 function in the epigenetic reprogramming of its binding sites, comparison of ChIP-seq data in mesodermal cells (1-day) and myogenic progenitors (6-day) for these histone marks revealed a time-dependent increase in H3K4me1 and H3K27Ac at 83% (3132/3780 total) of PAX3 binding sites (clusters 1A and 4A in Supplementary Fig. 3d, e).

To further demonstrate the ability of these elements to drive tissue-specific transcription in vivo, we took advantage of the VISTA enhancers catalog published by Visel and colleagues[27]. The VISTA catalog contains a list of annotated enhancers examined based on their ability to drive regional-specific LacZ expression in transgenic E11.5 embryos. Out of 52 enhancers annotated as active in the somites, 9 overlapped with PAX3 ChIP-seq peaks (Fig. 2d, f and Supplementary Fig. 5a–c). These include an element located ~30 kb from the gene *Gas1* (Fig. 2d, e), a Sonic Hedgehog inhibitor expressed in the neural tube and somite[28], and a PAX3 binding site within the second intron of the gene *Adamts5* (Fig. 2f, g), which is also expressed in developing muscles[29]. Overlapping of PAX3 binding sites was also observed for elements active in the developing limb and in the neural tube (Supplementary Fig. 5a, d), which represent known Pax3-expressing tissues[30]. Collectively, these data identify the compendium of PAX3-bound enhancers capable of driving tissue-specific gene expression during the development of embryonic mesoderm, and demonstrate that Pax3 regulates the deposition of histone marks commonly present at active enhancers and located many kilobases from the nearest TSS.

**Pax3 interacts with chromatin looping proteins**. To determine the molecular mechanisms behind Pax3 function, we purified the PAX3 complex and used mass spectrometry to define the PAX3 interactome. For this, we generated a Pax3-tagged cell line (iPax3HaFlag), which displayed equal myogenic activity as the untagged protein (Supplementary Fig. 6a, b). This HaFlag-fusion protein strategy enables tandem affinity purification (TAP) using anti-FLAG and anti-HA magnetic beads (Fig. 3a). When compared with the untagged Pax3 line, sequential purification of PAX3HaFlag resulted in the specific isolation of PAX3 (Fig. 3b) and its associated proteins, as estimated by silver staining (Fig. 3c). Two independent purifications were then subjected to mass spectrometry analysis to identify potential candidate cofactors (Supplementary Data 4). Upon elimination of the background (IP from untagged Pax3 line), proteins identified following this approach were analyzed using STRING to assess the presence of specific protein complexes (Supplementary Data 5). Interestingly, this analysis revealed the presence of proteins belonging to the histone methyltransferase (HMT) complex and several proteins involved in chromatin looping, including subunits of the Cohesin complex and LDB1 (Fig. 3d). The HMT complex has been identified in PAX2, PAX5, and PAX7 interactomes[31–33], where it mediates H3K4 methylation at their respective binding sites[33–36]. On the other hand, interaction between Pax factors and chromatin looping proteins has never been reported. The Cohesin complex plays a major role in higher order chromatin structure by cooperating with CTCF at contact domain boundaries. The LIM-domain protein LDB1 is a nuclear cofactor, which is required for long-range interactions between the locus control region (LCR) and the β-globin genes[37,38]. LDB1 is expressed broadly and its function goes beyond globin gene expression, as it also interacts with ISL1 in cardiac progenitors and pancreatic β cells[39,40] and LHX2 during olfactory receptor choice[18]. Co-immunoprecipitation followed by western blot confirmed interactions between PAX3 and LDB1, ASH2L and the Cohesin subunit SMC1 (Fig. 3e). Consistently, reciprocal co-immunoprecipitation using anti-LDB1 antibody confirmed interaction with PAX3 and the closely related myogenic transcription factor PAX7 (Supplementary Fig. 6c, d). Importantly, we observed co-localization of PAX3-LDB1 and PAX3-SMC1 in the nuclei of E9.5 mouse embryo somites (Fig. 3f and Supplementary Fig. 6e). Together, these data indicate the existence of PAX3 complexes, including LDB1 and the Cohesin subunit SMC1.

**Ldb1 is recruited at Pax3-bound enhancers**. To confirm that these newly identified chromatin looping proteins are bound to Pax3-regulated enhancers in mesodermal cells, we performed ChIP-seq using antibodies specific for LDB1 and Smc1. This

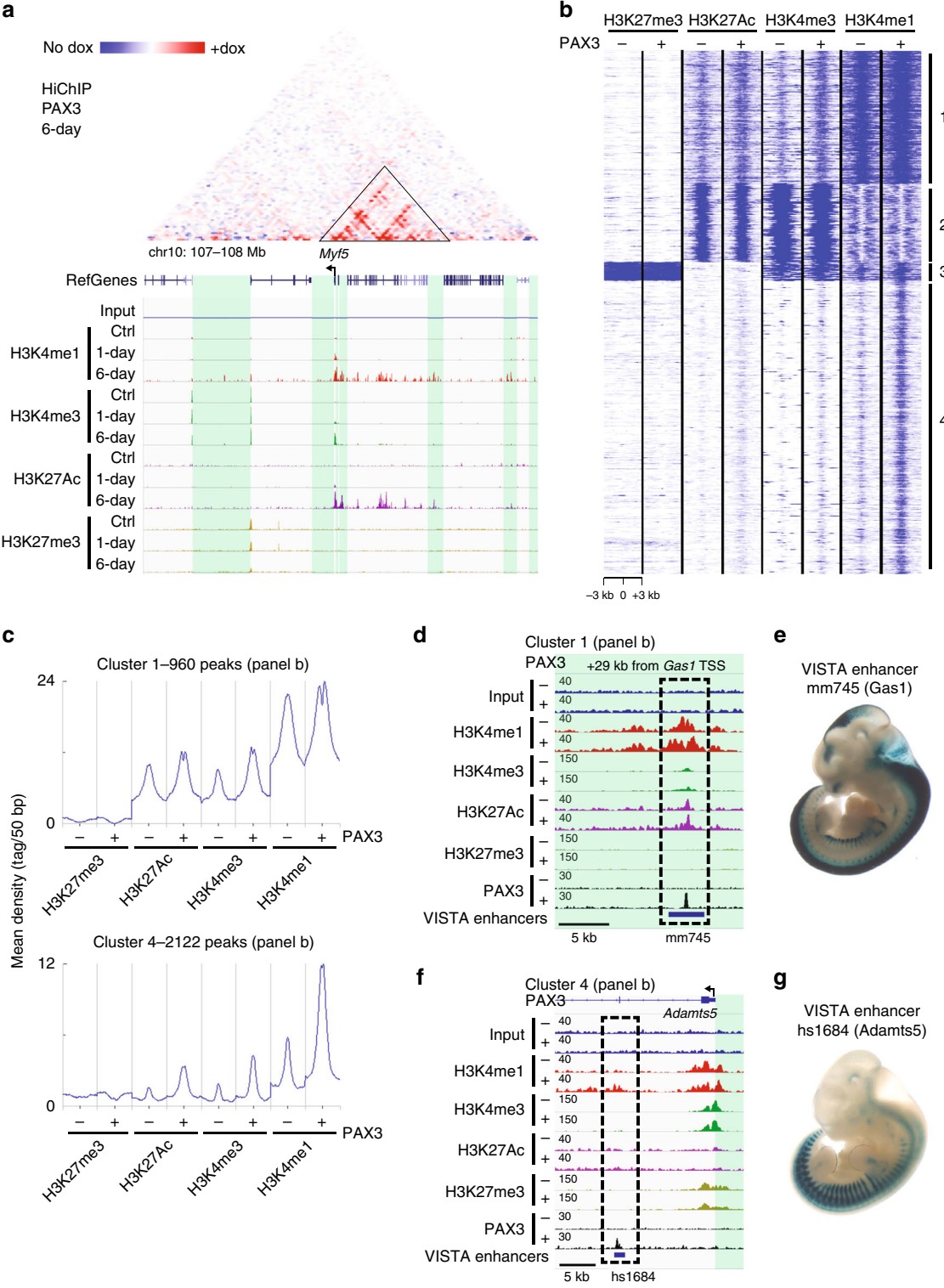

analysis yielded 4483 LDB1 and 13377 SMC1 peaks in Pax3-induced mesodermal cells (1-day), whereas non-induced displayed 6317 LDB1 and 19082 SMC1 peaks (Supplementary Fig. 7a and Supplementary Data 3). Although not detected by mass spectrometry, we also analyzed CTCF chromatin binding since the CTCF/SMC1 complex is necessary for topologically associated domain formation[6–8]. This analysis yielded 55177 and 53669 CTCF peaks in 1-day Pax3-induced and non-induced cells, respectively (Supplementary Fig. 7a and Supplementary Data 3). Comparison of these ChIP-seq data demonstrated a high degree

of overlap between SMC1 and CTCF peaks (92% and 87% in dox-induced and non-induced iPax3 cells, respectively; Supplementary Fig. 7a). On the other hand, only a small percentage of LDB1 sites overlapped with SMC1 and CTCF peaks (22% and 25% in dox-induced and non-induced iPax3 cells, respectively; Supplementary Fig. 7a). As shown in Fig. 4a, k-means clustering detected LDB1 and SMC1 binding to a subset of PAX3 sites (clusters 1, 2, and 3). Whereas SMC1 appeared to be bound independently of PAX3 induction (cluster 3 in Fig. 4a), this was not the case for LDB1 as PAX3-dependent LDB1 recruitment at

**Fig. 2** Epigenetic reprogramming of PAX3 binding sites. **a** HiChIP normalized matrix from 6-day Pax3-induced cells displaying the *Myf5* locus shows that the region characterizing increased contacts (triangle) undergoes changes in deposition of H3K4me1 and H3K27Ac marks over time. Scale: maxrange = 0.2. IGV snapshot displaying the ChIP-seq tracks for H3K27Ac, H3K27me3, H3K4me3, and H3K4me1 in non-induced, 1-day and 6-day induced iPax cells is shown below. Green boxes represent intergenic regions. Black arrow represents the transcription start site. **b** k-means clustering of H3K4me1, H3K4me3, H3K27me3, and H3K27Ac ChIP-seq data from 1-day Pax3-induced (+) and non-induced (−) EB cultures. Mapped data were used to generate a Density Tag Map centered on 3780 PAX3 peaks ±3 kb. Cluster 1: 960 peaks; Cluster 2: 566 peaks; Cluster 3: 136 peaks; Cluster 4: 2122 peaks. **c** Distribution of H3K4me1, H3K4me3, H3K27me3, and H3K27Ac ChIP-seq reads across the PAX3 peak center ±3 kb for clusters 1 and 4 shown in panel **b**. **d–f** IGV track displaying H3K4me1, H3K4me3, H3K27Ac, H3K27me3, and PAX3 genomic occupancy at representative PAX3-bound loci from clusters 1 (*Gas1*) and 4 (*Adamts5*) in 1-day Pax3-induced (+) and non-induced (−) EBs cultures. Dashed black squares indicate PAX3-bound loci. Green boxes represent intergenic regions. Black arrow represents the transcription start site. **e–g** Images of E11.5 transgenic embryos expressing LacZ under the control of VISTA elements overlapping to Pax3 peaks. mm745, adjacent to the *Gas1* locus, is expressed in both somite and neural tube. hs1684, within *Adamts5* gene, is expressed in the somite. Published images were downloaded from https://enhancer.lbl.gov

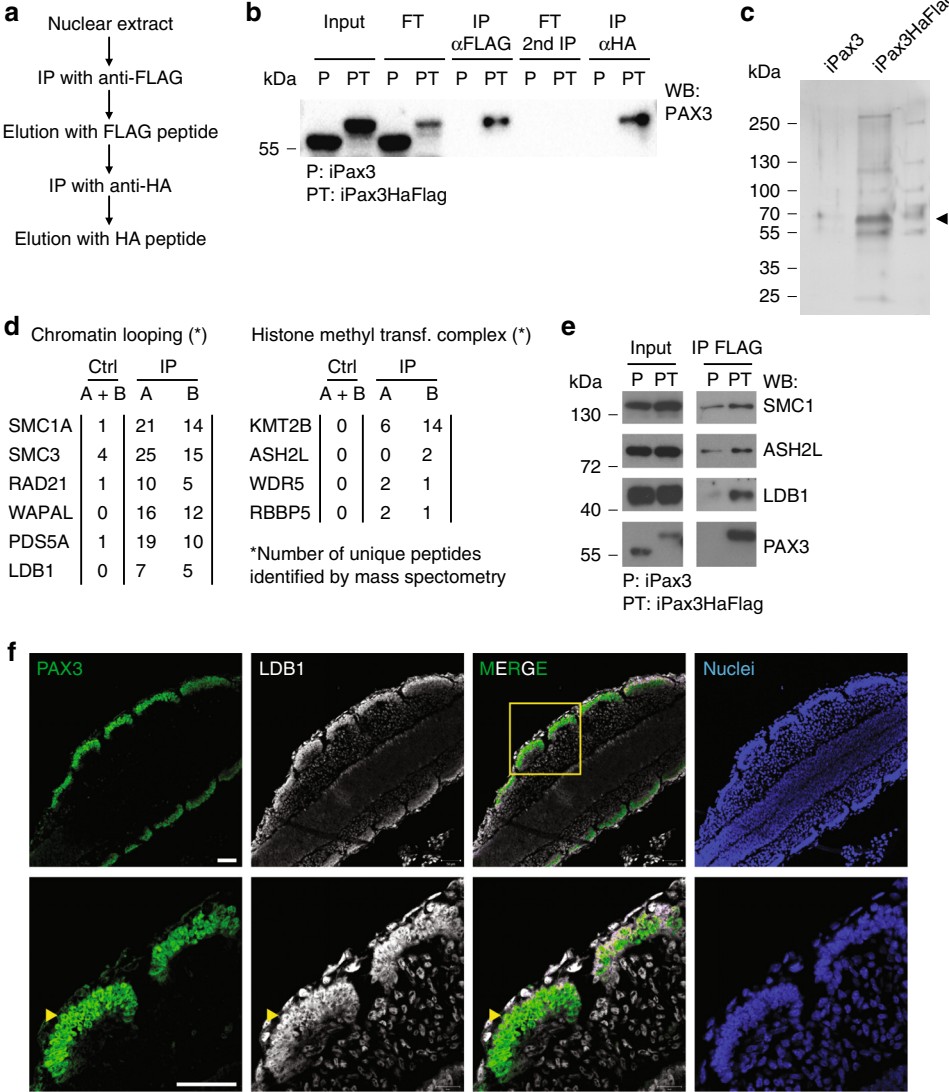

**Fig. 3** Identification of the PAX3 protein complex by mass spectrometry. **a** Scheme of tandem affinity purification of PAX3. **b** Western blot using anti-PAX3 antibody. **c** Silver staining of TAP samples from iPax3 and iPax3HaFlag 1-day induced EBs. **d** Unique peptides detected for proteins associated to chromatin looping and histone methylation. **e** Validation of PAX3 interaction with LDB1, ASH2L, and SMC1 in 1-day induced EBs. Images are representative of independent biological replicates (n = 3). **f** Immunofluorescence staining shows co-localization of PAX3 and LDB1 in the somites of E9.5 embryos. Upper panel: ×20 magnification. Lower panel ×63 magnification. PAX3 (green); LDB1 (white); Nuclei (blue). Scale bar: 50 μm. Source data are provided as a Source Data file

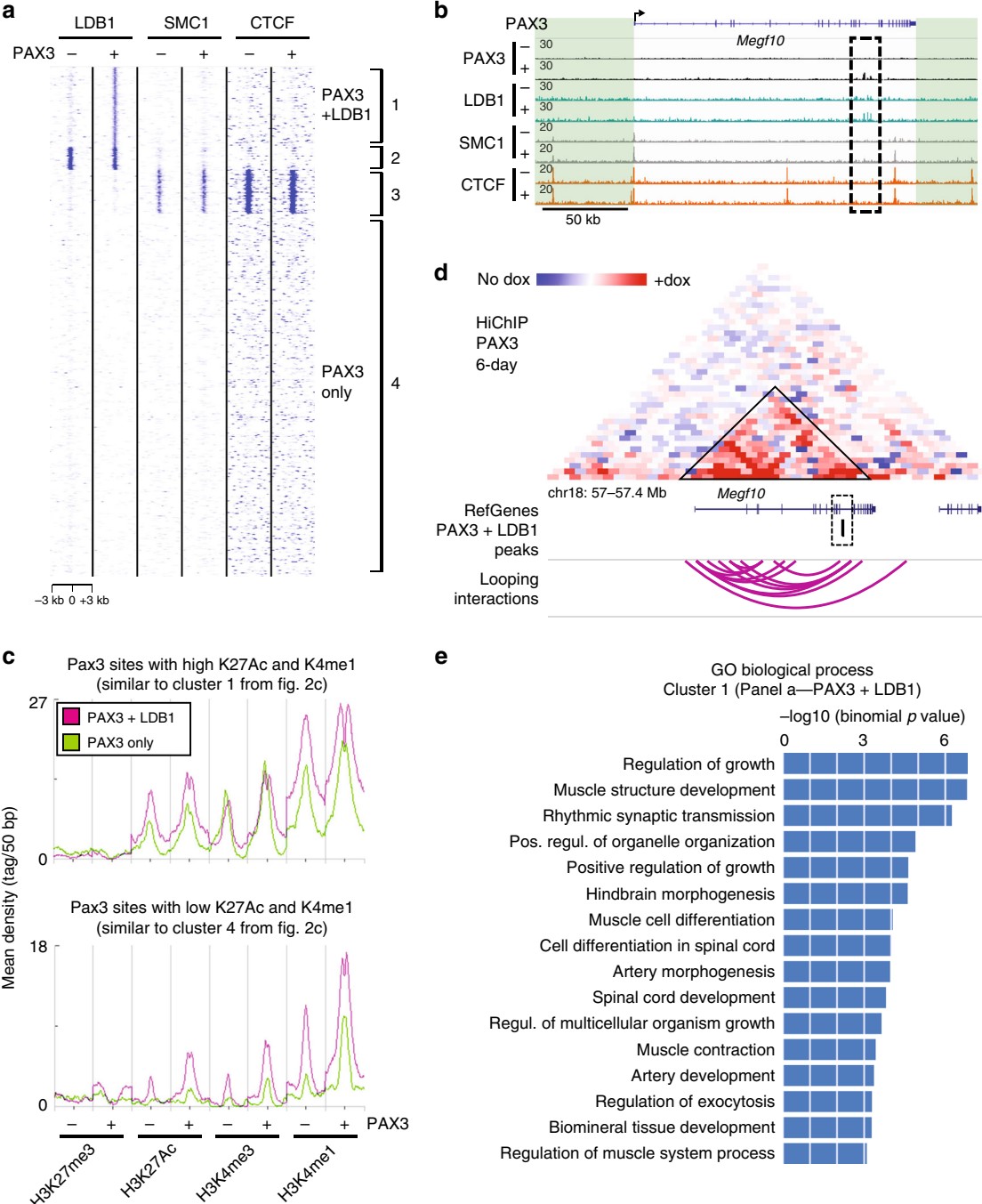

**Fig. 4** Ldb1 recruitment at Pax3 sites is associated with enhanced chromatin remodeling and looping. **a** k-means clustering of ChIP-seq data for LDB1, SMC1, and CTCF in 1-day Pax3-induced (+) and non-induced (−) EBs. Density Tag Map centered on 3780 PAX3 peaks shows the recruitment of LDB1 upon Pax3 induction at a subset of loci (cluster 1), and reveals that PAX3-mediated LDB1 recruitment is independent from SMC1-CTCF complex. Cluster 1: 595 peaks; Cluster 2: 168 peaks; Cluster 3: 324 peaks; Cluster 4: 2693 peaks. **b** IGV track displaying PAX3, LDB1, SMC1, and CTCF genome occupancy at the *Megf10* locus in 1-day Pax3-induced (+) and non-induced (−) EBs. Dashed black square indicate PAX3-mediated LDB1 recruitment. Green boxes represent intergenic regions. Black arrow represents the transcription start site. **c** Graph shows distribution of ChIP-seq reads for the selected marks at loci characterized by PAX3 + LDB1 recruitment (violet) and PAX3-only binding (green). Loci bound by both LDB1 and PAX3 display higher remodeling compared with PAX3-only bound sites. **d** HiChIP normalized matrix from 6-day Pax3-induced cells displaying the *Megf10* locus demonstrates long-range interactions involving the 1-day bound PAX3 + LDB1 site (dashed black square from panel **b**). Position of PAX3 peaks, genes and chromosome coordinates are showed below the matrix. Scale: maxrange = 0.2. Arcs indicate looping interactions identified by FitHiChIP. **e** GREAT functional annotation of peaks characterized by PAX3-mediated LDB1 recruitment (cluster from Fig. 4a)

cluster 1 loci was evident (Fig. 4a, b and Supplementary Fig. 7e), which we confirmed by ChIP-qPCR (Supplementary Figs. 7b and 8a and see below). This analysis identified 595 loci characterized by PAX3-dependent LDB1 recruitment (hereafter referred as PAX3 + LDB1 sites). Following annotation to the nearest TSS, this last group of loci includes genes essential for myogenesis, such as *Eya1* and *Fgfr4*, as well as multiple differentially regulated genes based on RNA-seq data (185/539 gene annotated peaks; Supplementary Fig. 7c). As observed for SMC1, LDB1 binding to chromatin was also detected at a subset of sites independent of PAX3 expression (cluster 2 in Fig. 4a), suggesting that at some sites (clusters 2 and 3), LDB1 and SMC1 are present before PAX3 occupancy. In the case of SMC1, CTCF-dependent Cohesin recruitment is well-established, and accordingly, we detected enrichment for CTCF at the same PAX3 sites bound by SMC1 (cluster 3 in Fig. 4a). These results were consistent across biological replicates (Supplementary Fig. 4b). Importantly, our data show that PAX3 bound elements are occupied either by LDB1 or SMC1/CTCF but not both (Fig. 4a, b and Supplementary Fig. 7e), implying a differential role for these proteins in the regulation of Pax3-mediated myogenic activation.

The evident PAX3-dependent LDB1 recruitment observed at cluster 1 loci prompted us to deeply investigate the function of Ldb1 in the activation of the myogenic program. As not all PAX3 sites displayed LDB1 recruitment, we compared the deposition of histone marks between these two groups of loci by taking advantage of the 1-day ChIP-seq data for H3K27me3, H3K27Ac, H3K4me1, and H3K4me3 (Fig. 2c). To avoid biases due to group size, we used an equal number of randomly selected Pax3 sites for the two groups. Clustering of the histone ChIP-seq reads on a region spanning ±3 kb from the center of the PAX3 peaks demonstrated a distribution similar to the one observed in Fig. 2c for both PAX3 + LDB1 and PAX3 only loci. However, loci co-occupied by PAX3 and LDB1 displayed a more pronounced bimodal distribution and deposition of the H3K27Ac and H3K4me1 marks compared with the loci bound by PAX3 only (Fig. 4c). This was clear in both clusters characterized by H3K4me1 + low and high H3K27Ac levels (Fig. 4c), and was consistent across biological replicates (Supplementary Fig. 4c). The differential function of PAX3 + LDB1 and PAX3 only loci is supported by the Gene Ontology annotation of the genes associated with these elements. Using GREAT[41], we observed that PAX3 + LDB1 elements were associated with genes involved in muscle development and embryonic patterning (Fig. 4e). In contrast, PAX3 only loci displayed an enrichment for a more heterogeneous group of biological process categories (Supplementary Fig. 7g).

Looking into our chromatin looping data, we observed that PAX3 + LDB1 sites from our 1-day ChIP-seq analysis overlap with looping anchors detected in our 1-day and 6-day HiChIP studies, respectively. Examples of such interactions involve the *Megf10* + 135 kb and *Prex1* + 158 kb elements characterized by PAX3-mediated LDB1 recruitment (Fig. 4b, d and Supplementary Fig. 7e, f). Based on these results, we believe that Ldb1 recruitment at a subset of Pax3 sites may play an important role in both deposition of epigenetic marks, and ultimately, chromatin looping of the enhancers associated with the activation of the myogenic program.

**Myogenic differentiation is impaired upon Ldb1 knockdown.** Recruitment of Ldb1 occurs at a subset of Pax3 sites overlapping with in vivo functional enhancers characterized by epigenetic remodeling (Fig. 2e and Supplementary Fig. 7d). Importantly, these PAX3 + LDB1 sites are associated with genes involved in mesoderm development (Fig. 4e), including known regulators of

the myogenic lineage, such as *Cdon*, *Eya1*, *Fgfr4*, and *Megf10*[42–45]. Since a role for Ldb1 during skeletal myogenesis has never been investigated, we next determined the functional consequences of reducing Ldb1 levels using RNA interference in Pax3-inducible ES cells. As shown by western blot, immunostaining, and gene expression analyses (Fig. 5a–c), the knockdown of Ldb1 dramatically impaired the ability of Pax3 to induce the skeletal myogenic program.

**Ldb1 regulates chromatin looping and H3K4me1 deposition.** We then investigated how epigenetic modification of the PAX3-bound enhancers is affected upon Ldb1 knockdown. Because our ChIP-seq data for histone marks indicated that deposition of H3K4me1 and H3K27Ac marks was more pronounced at PAX3 + LDB1 compared with PAX3 only sites, we tested whether lack of Ldb1 would affect this observation. Our results showed this to be the case as Ldb1 knockdown resulted in decreased deposition of the H3K4me1 mark at the sites occupied by PAX3 and LDB1 (Fig. 5d). Using PAX3 antibody, we performed HiChIP in Ldb1 knockdown (shLdb1) and control (shSCR) Pax3-induced myogenic progenitors. In independent replicates, we detected fewer significant contacts upon Ldb1 knockdown (Fig. 5e), which resulted in diminished numbers of chromatin loops (9565 and 6885 in control vs. Ldb1 knockdown, respectively; Fig. 5f and Supplementary Data 1). Visual inspection of the HiChIP matrices confirmed the reduced long-range interactions at loci characterized by PAX3-dependent LDB1 recruitment (*Dbx1* and *Prex1* loci in Fig. 5g and *Cdon* in Supplementary Fig. 7h). We conclude that Ldb1 participates in the Pax3-mediated myogenic specification process by enabling efficient posttranslational modification of histone 3 tails at Pax3 sites and chromatin looping of these enhancers.

**Forced Ldb1 targeting to Pax3 sites induces myogenesis.** Based on our proteomic, epigenetic, and transcriptomic data, Pax3 specifies the skeletal myogenic program by cooperating with Ldb1 and the HMT complex (Fig. 6a). Nonetheless, a clear structure-function understanding of the PAX3 interactome requires the analysis of each individual protein in a controlled experimental setting. The Pax3-inducible ES cell system represents the ideal tool for these studies as derivation of skeletal muscle cells is entirely dependent on Pax3 induction (Supplementary Fig. 1a–d). We have previously reported that a Pax3-deletion mutant lacking the C-terminal transactivation domain (TAD) region (also referred as ΔTAD; Fig. 6b) is devoid of myogenic activity[46] (Fig. 6d). Importantly, we show here that the ΔTAD mutant is incapable of recruiting LDB1 nor inducing H3K4me1 deposition at PAX3 binding sites (Fig. 6c and Supplementary Fig. 8a, e), while still maintaining its DNA binding capability (Supplementary Fig. 8b). Therefore, we took advantage of this Pax3 mutant to investigate the requirement of individual candidate interacting proteins in the Pax3-mediated skeletal myogenic commitment. We generated dox-inducible ES cell lines expressing ΔTAD fusion proteins in frame with the cDNA encoding for Ldb1, Ash2l, Cxxc1, and Pagr1a (Fig. 6b and Supplementary Fig. 8c). ASH2L is a common subunit of the HMT complex, while CXXC1 (also called CFP1) and PAGR1A (also known as PA1) have been associated with the HMT complexes responsible for H3K4me3 and H3K4me1 deposition, respectively[47,48]. Although CXXC1 and PAGR1A were not detected in our mass spectrometry, we included these two proteins here for a more comprehensive screening. As expected, Pax3 induction (WT) resulted in robust myogenic differentiation, as measured by MYOG immunostaining on terminally differentiated

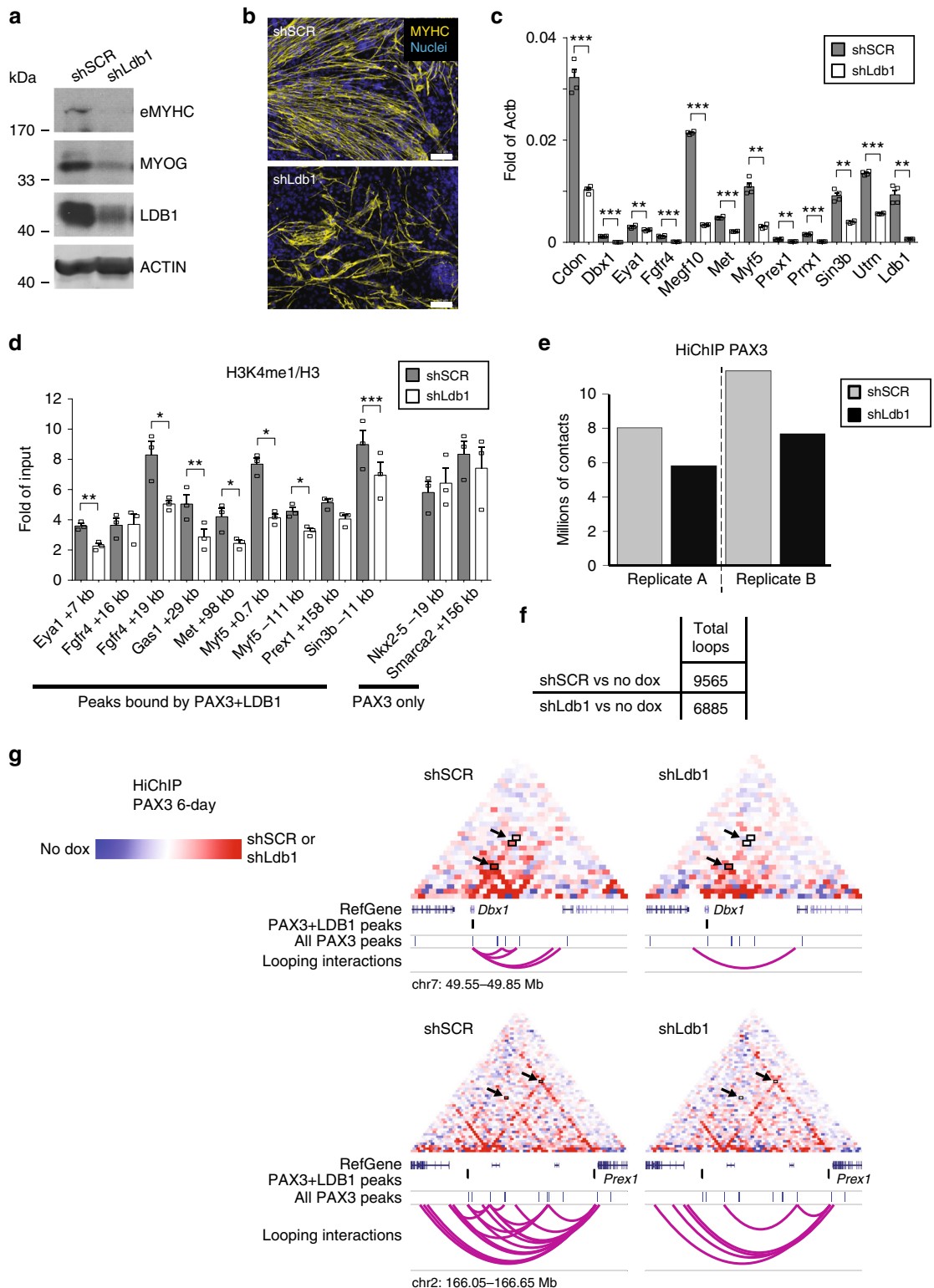

cultures from 6-day dox-induced cells (Fig. 6d). Interestingly, among the fusion proteins, only ΔTAD-LDB1 was capable of efficiently inducing the myogenic program (Fig. 6d). This was consistent with gene expression data of mesodermal cells (1-day induced), as most Pax3 target genes are efficiently upregulated only by WT and ΔTAD-LDB1 proteins (Fig. 6e and Supplementary Fig. 8d). To investigate the mechanism underlying transcriptional activation, we performed HiChIP in non-induced (no dox) and 1-day dox-induced (+dox) ΔTAD-

LDB1 cell cultures. This analysis detected 2798 looping interactions induced by ΔTAD-LDB1 within 24 h (Supplementary Data 1) and, as shown in Fig. 6f, several of these chromatin looping events occurred at the genomic loci associated with ΔTAD-LDB1 upregulated genes (*Cdon*, *Dbx1*, *Myf5*, and *Prrx1*). Based on these results, we conclude that forced recruitment of Ldb1 to Pax3 sites mediates the formation of long-range interactions responsible for gene activation.

**Fig. 5** Myogenic differentiation and looping are impaired upon Ldb1 knockdown. **a–c** Ldb1 knockdown in iPax3 differentiating cells impairs myogenic differentiation. Western blot (**a**) shows reduced levels of eMYHC, MYOG, and LDB1 in differentiated cultures from 6-day Pax3-induced Ldb1-knockdown (shLdb1) and control (shSCR) ES cells. Similar results were obtained by immunostaining using eMYHC (**b**) and gene expression analysis of Pax3-regulated genes. Scale bar: 100 μm (**c**). Graph represents mean + s.e.m of $n \geq 3$ independent biological replicates. Student's $t$-test $**p < 0.01$, $***p < 0.001$. **d** Knockdown of Ldb1 in iPax3 cells (collected upon 6-day Pax3 induction) impairs the deposition of H3K4me1 at PAX3 + LDB1 sites. Graph represents mean + s.e.m. of $n = 3$ independent biological replicates. Student's $t$-test $*p < 0.05$, $**p < 0.01$, $***p < 0.001$. **e** Ldb1 knockdown impairs chromatin looping at Pax3 sites. Graph shows the number of contacts detected at PAX3 + LDB1 sites. **f** Number of long-range interactions (loops) detected by Pax3 HiChIP in shSCR and shLdb1 cells. **g** HiChIP normalized matrix from shSCR and shLdb1 6-day Pax3-induced cells displaying the *Dbx1* and *Prex1* loci. The contact maps were normalized for sequencing depth and visualized as ratio relative to the background (HiChIP in non-induced cells). Black arrows indicate the loss in interaction frequency between two loci. Position of PAX3 peaks, genes, and chromosome coordinates are showed below the matrix. Scale: maxrange = 0.2. Arcs indicate looping interactions identified by FitHiChIP. Source data are provided as a Source Data file

Since Pax3 expression induces deposition of H3K4me1 at a subset of enhancers (Fig. 2b) and co-immunoprecipitation using LDB1 antibody shows interaction with PAX3, ASH2L, and WDR5 (Supplemental Fig. 5c), we next investigated the effect of ΔTAD-LDB1 and ΔTAD-ASH2L proteins on histone 3 posttranslational modification. Consistently, ChIP using the H3K4me1 antibody demonstrated increased levels of this mark at PAX3 bound elements (e.g., *Myf5* −111 kb enhancer) following induction of full length PAX3, whereas the ΔTAD mutant lacked this activity (Supplemental Fig. 7e). Similar results were obtained following induction of ΔTAD-LDB1, whereas ΔTAD-ASH2L was able to induce H3K4me1 only at a subset of Pax3 sites (Supplementary Fig. 8e). To verify that ΔTAD-LDB1 function is not restricted to a few Pax3 loci, we performed ChIP-seq for H3K4me1 in 1-day dox-induced and non-induced ΔTAD and ΔTAD-LDB1 mesodermal cells. Inspection of genomic tracks for dox-induced and non-induced WT, ΔTAD, and ΔTAD-LDB1 lines confirmed our ChIP-qPCR data at important myogenic loci (*Met*, *Myf5* −111 kb and *Gas1*; Supplementary Fig. 8f). On a global scale, using the list of 1-day PAX3 bound elements, k-means clustering identified that only a subset of sites (3725/7508) display a ΔTAD-LDB1-dependent increase in H3K4me1 levels (Fig. 6g, h and Supplementary Fig. 8g). Altogether, these data highlight the necessity of LDB1 recruitment at PAX3 bound elements for the proper activation of the myogenic program, and suggest that recruitment of the HMT complex alone is not sufficient for inducing gene activation.

**Ldb1 is required for efficient hypaxial myogenesis.** One of the most recognized functions of Pax3 during development is the regulation of myogenic progenitor migration toward the limb bud[49]. *Ldb1*-null alleles are embryonic lethal and mutants do not develop beyond gastrulation[50], therefore we used an *Ldb1* flox allele previously generated by the Westphal group[51]. To study Ldb1 function specifically during muscle development, we deleted Ldb1 in the Pax3 lineage by crossing *Ldb1^flox* and *Pax3^cre* mice[30]. We confirmed deletion by immunostaining for LDB1 in E9.5 embryos (Supplementary Fig. 9a). Myogenic progenitor migration to the forelimb bud is normally observed at E10.5. We found that control embryos (*Pax3^cre/+;Ldb1^fl/+*) displayed extensive migration at this time point, but Ldb1-deleted littermates (*Pax3^cre/+;Ldb1^fl/fl*) had severely reduced numbers of PAX3⁺ migrating progenitors, and most of those that were observed did not express MYF5 (Fig. 7a, b). Similar results were obtained in E11.5 control and Ldb1-deleted embryos using the myogenic marker MYHC (Fig. 7c, d). Based on these data, we conclude that Ldb1 is required for proper activation of the transcriptional program regulating the commitment and the migration of PAX3⁺ myogenic progenitors.

## Discussion

Here we demonstrate that lineage specification involves long-range interactions coordinated by a tissue-specific transcription factor. Gene expression relies on a complex interplay between promoters and enhancers, with the latter representing a major determinant of tissue specificity. Although chromatin looping enables the establishment of enhancer–promoter interactions, the role of transcription factors in shaping the three-dimensional organization of the genome during cell differentiation has been investigated in the context of only a few isolated loci[52–54]. To date, most of the genome-wide studies have focused on general mechanisms involving CTCF-[3,4,7,55] and YY1-dependent genome organization[15]. Nonetheless, recent studies investigating the three-dimensional rearrangements occurring during cell differentiation demonstrated a correlation between TF-binding and enhancer loop formation[16–18]. Our genome-wide analyses pinpoint how Pax3 orchestrates the skeletal myogenic program by promoting remodeling of the epigenetic landscape and chromatin looping of the myogenic loci (Figs. 1 and 2). We demonstrate that Pax3 activity relies on the recruitment of the chromatin looping factor Ldb1 (Fig. 3), which is required for establishing long-range interactions (Fig. 5) and, ultimately, the proper execution of the myogenic program during embryonic development (Figs. 5 and 7). In support of the instrumental role for Ldb1 during Pax3-dependent transcriptional activity, we show that targeting LDB1 to PAX3 bound sites using a PAX3 fusion protein (ΔTAD-LDB1) recapitulated Pax3 pro-myogenic function (Fig. 6d) and induced chromatin looping (Fig. 6f). Interestingly, this effect was specific for LDB1 as fusion proteins involving subunits of the HMT complex failed to robustly induce myogenesis, regardless of their ability to increase H3K4me1 levels at a subset of Pax3 sites (Supplementary Fig. 8e).

Our data elicits an unexpected correlation between Ldb1 and posttranslational modification of the histone 3 tails by the HMT complex. LDB1-recruitment to PAX3 binding sites is associated with increased deposition of H3K4me1 (Fig. 4c), and upon Ldb1 knockdown, the levels of this mark are diminished at PAX3-bound elements (Fig. 5d). Co-immunoprecipitation of LDB1 in iPax3 cells showed the interaction with the HMT complex sub-units ASH2L and WDR5 (Supplementary Fig. 6c), and expression of the PAX3 ΔTAD-LDB1 fusion protein was able to induce an increase in H3K4me1 at a subset of PAX3 bound sites (Fig. 6 and Supplementary Fig. 8). Since chromatin looping (and gene expression) is reduced upon Ldb1 knockdown (Fig. 5e–g), we speculate the recruitment of the HMT complex may be affected by the local chromatin conformation of the PAX3-LDB1-bound sites. These elements are enriched for genes involved in mesoderm development (Fig. 4e), and we predict they may represent important enhancers for this transcriptional program. Accordingly, LDB1 occupancy has been demonstrated at several erythroid enhancers, which positively correlated with the expression levels of the respective genes[56].

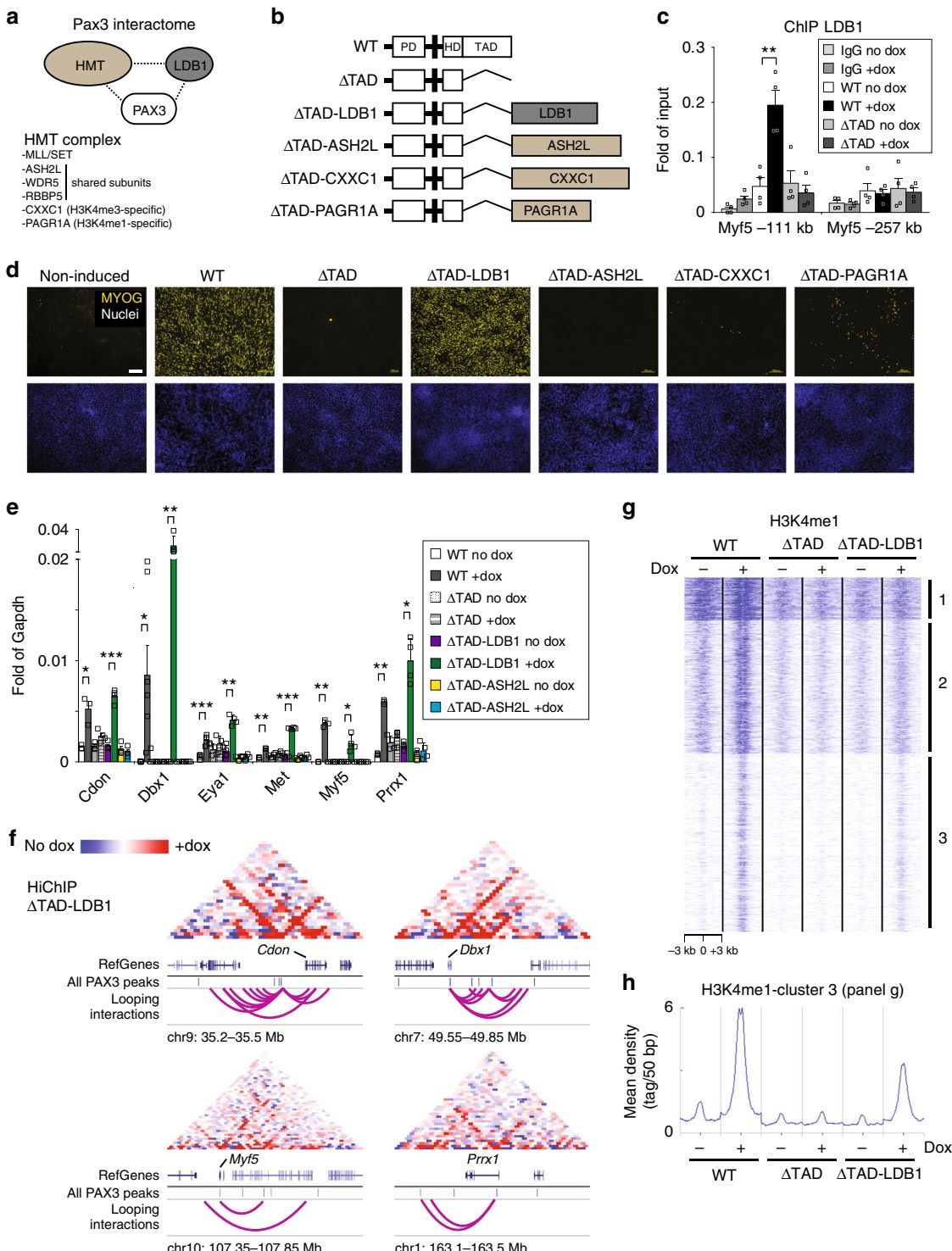

Previous studies in hematopoietic cells have shown that Ldb1 determines β-globin expression by promoting the interaction between the LCR and the β-globin gene promoters[38]. Ldb1 regulates this process by establishing chromatin looping independently of the Mediator and Cohesin complexes, and by increasing the frequency of transcription bursts[57,58]. In our results, the overlap between LDB1-bound and SMC1/CTCF-bound PAX3 loci is limited (Fig. 4a), thus suggesting that chromatin looping involving PAX3-LDB1 sites does not recruit the Cohesin complex. Nevertheless, a recent study demonstrated that chromatin looping can also occur between LDB1 and CTCF-bound loci[59].

Taking these data in consideration, our findings raise the intriguing hypothesis that Ldb1 may be one of the key players in the establishment of the sub-topologically associated domain interactions observed during cell differentiation, thus complementing the function of CTCF-Cohesin. Since LDB1 can homodimerize[60], it is possible that direct LDB1 interactions contribute to nucleation of the chromatin domains centered on PAX3 bound sites (Figs. 1 and 4). In agreement with this model, our data using ΔTAD-LDB1 (Fig. 6f), and a previous report by the Blobel group[37], indicate that forced recruitment of Ldb1 is sufficient to mediate chromatin looping and induce/rescue transcription.

**Fig. 6** Forced LDB1 targeting to PAX3 bound elements induces skeletal myogenesis. **a** Schematic representation of PAX3 interactions with the HMT complex and LDB1. CXXC1 and PAGR1A subunits are present in the HMT complex responsible for deposition of H4K4me3 (SET1A complex) and H4K4me1 (MLL3/4 complex), respectively. **b** Schematic representation of PAX3 (WT), PAX3 lacking the transactivation domain (ΔTAD) and the fusion proteins involving LDB1, ASH2L, CXXC1, and PAGR1A. Size is not indicative of the real protein length. Pax3 domains: PD, paired domain; HD, homeodomain; TAD, transactivation domain. **c** The PAX3 TAD is required for LDB1 recruitment. ChIP-qPCR analysis of LDB1 recruitment at the *Myf5* −111 kb locus in 1-day WT- and ΔTAD-induced (+) and non-induced (−) cells. *Myf5* −257 kb: negative control. Graph represents mean + s.e.m. from $n = 4$ independent experiments. Student's *t*-test **$p < 0.01$. **d** Analysis of the skeletal myogenic activity of WT, ΔTAD, ΔTAD-LDB1, ΔTAD-ASH2L, ΔTAD-CXXC1, and ΔTAD-PAGR1A. Transgenes were induced for 6-day following the scheme presented in Fig. 1a. Non-induced cells are used as negative control of myogenic commitment. Immunostaining: MYOG antibody (yellow) DAPI (Nuclei - blue). Images are representative of independent biological replicates ($n = 3$). Scale bar: 100 μm. **e** Gene expression analysis of Pax3 targets following 1-day induction of WT, ΔTAD, ΔTAD-LDB1 and ΔTAD-ASH2L. Graph represents mean + s.e.m. of $n \geq 3$ independent biological replicates. Student's *t*-test *$p < 0.05$, **$p < 0.01$, ***$p < 0.001$. **f** Long-range interactions at the *Cdon*, *Dbx1*, *Myf5*, and *Prrx1* loci in 1-day ΔTAD-LDB1-induced cultures. Normalized contact maps based on sequencing depth are visualized as ratio relative to the non-induced cells. Position of PAX3 peaks, genes, and chromosome coordinates are displayed below the matrix. Scale: maxrange = 0.2. Arcs indicate looping interactions identified by FitHiChIP. **g** k-means clustering of H3K4me1 ChIP-seq data from 1-day induced (+) and non-induced (−) WT, ΔTAD, and ΔTAD-LDB1 cultures. Density Tag Map for H3K4me1 is centered on 3780 PAX3 peaks ±3 kb. Cluster 3 shows an increase in H3K4me1 deposition following WT and ΔTAD-LDB1 induction. Cluster 1: 457 peaks; Cluster 2: 1420 peaks; Cluster 3: 1903 peaks. **h** Distribution of H3K4me1 ChIP-seq reads across the PAX3 peak center ±3 kb for cluster 3 shown in panel **g**. Source data are provided as a Source Data file

Taken together, our findings demonstrate that PAX3 bound loci engage in LDB1-dependent long-range interactions and support the contribution of Ldb1 in the efficient chromatin remodeling of these elements (Fig. 7e), which is instrumental for the proper execution of the hypaxial myogenic gene expression program.

## Methods

**Plasmids**. The p2lox-Pax3 and p2lox-Pax3-ΔTAD-3xFLAG vectors were previously described[46]. The p2lox-Pax3-HaFlag vector was generated by cloning the HaFlag sequence (synthetized as G-block—IDT) into the NotI site of the p2lox-Pax3-noStopCodon (described in ref. [46]). The HaFlag G-block sequence is provided in Supplementary Table 1. The mouse cDNAs encoding for Ash2l (accession number BC012957—Dharmacon), Cxxc1 (accession number BC030938—Dharmacon), Ldb1 (accession number BC013624—Dharmacon) and Pagr1a (accession number NM_030240.1) were amplified by PCR and cloned 3′ relative to the Pax3 homeodomain. Cloning details and plasmid maps are available upon request. Knockdown constructs from the TRC library (pLKO) were obtained from the University of Minnesota Genomic Center (UMGC). pLKO.1-blast-SCRAMBLE was a gift from Keith Mostov (Addgene plasmid # 26701[61]). To purify cells transduced with the knockdown constructs, all pLKO plasmids were modified by replacing the selection gene (puromycin or blasticidin) with the sequence encoding the EGFP (referred hereafter as pLKO-shRNA-PGK-GFP).

**Cell cultures**. Inducible mES cell lines were generated by Cre-loxP mediated recombination of the p2lox targeting plasmids into A2lox-cre mouse ES cells[62]. The recombination cassette, located next to the Hprt gene, contains the tet-responsive-element (TRE) driving the expression of one single copy of cDNA, thus ensuring quasi-physiological expression levels. mES cells were maintained on mitotically impaired mouse embryonic fibroblasts (MEFs) in knock-out DMEM (Invitrogen) supplemented with 15% FBS (Embryomax ES-qualified FBS—Millipore), 1% penicillin/streptomycin (Invitrogen), 2 mM Glutamax (Invitrogen), 0.1 mM non-essential aminoacids (Invitrogen), 0.1 mM β-mercaptoethanol (Invitrogen), and 1000 U/ml LIF (Millipore). A detailed version of the myogenic differentiation protocol has been described[63]. Briefly, the ES/MEF cell suspension was preplated in a gelatin-coated dish for 30 min in order to remove fibroblasts and the resulting supernatant (enriched for mES cells) was then diluted to 40,000 cells/ml in EB differentiation medium and incubated in an orbital shaker at 80 RPM. EB differentiation medium: IMDM (Invitrogen) supplemented with 15% FBS (Embryomax ES-qualified FBS), 1% penicillin/streptomycin (Invitrogen), 2 mM Glutamax (Invitrogen), 50 μg/ml ascorbic acid (Sigma-Aldrich), 4.5 mM mono-thioglycerol (MP biomedicals). Transgene induction was achieved by adding doxycycline (Sigma-Aldrich) to day-3 EBs cultures (final concentration 1 μg/ml), and then maintained throughout the differentiation protocol by replacing the media (including dox) every 2 days. At day 5, EBs were disgregated and single cells were incubated for 20 min with PDGFRα-PE and FLK1-APC conjugated antibodies (e-Bioscience). PDGFRα + FLK1− cells were sorted using FACSAriaII (BD biosciences) and replated on gelatin-coated dishes using EB differentiation media supplemented with 1 μg/ml doxycycline and 10 ng/ml mouse basic-FGF (Preprotech). Skeletal myogenic differentiation was assessed in cells cultured 4 days (equivalent to 6-day Pax3 induction) as monolayer by withdrawing bFGF and doxycycline from the cultures (in order to shutdown transgene expression) followed by additional 2 days culture in the same serum- or serum-free media. HEK293T cells were maintained in DMEM (Invitrogen) supplemented with 10%

FBS (Millipore), 1% penicillin/streptomycin (Invitrogen) and 2 mM Glutamax (Invitrogen).

Intracellular staining using MYOD and MYOG antibodies was performed on non-induced (day-4 EBs), 1-day and 6-day Pax3-induced cells. Cells were collected and fixed with 1% paraformaldehyde/PBS for 15 min. Fixed cells were then incubated on ice for 1 h with antibodies diluted in 0.3% Triton X-100/0.5% BSA/PBS solution. The list of antibodies used in this study and relative dilutions are provided in Supplementary Table 2. After washing with 0.3% Triton X-100/0.5% BSA/PBS solution, cells were incubated on ice for 30 min with secondary antibodies (Alexa555 anti-mouse) in 0.3% Triton X-100/0.5% BSA/PBS solution. After washing with 0.3% Triton X-100/0.5% BSA/PBS solution, cells were analyzed with FACSAriaII.

**Lentiviral transduction and knockdown**. Lentiviruses were produced in HEK293T cells by co-transfection of pVSV-G, Δ8.9 and lentiviral constructs using Lipofectamine LTX-Plus reagent (Invitrogen). Supernatants containing lentiviral particles were filtered using 0.45 μm filters and applied to cells cultured in 6-well plates. To facilitate transduction, 6-well plates were centrifuged for 90 min at $1100 \times g$, 30 °C. Ldb1 knockdown was achieved by transduction of day-5 PDGFRα + FLK1− sorted cells with shSCR and shLdb1 lentiviral particles. Transduction efficiency (usually between 95 and 99% GFP + cells) was assessed 3 days post-transduction using FACS. Replated day-5 PDGFRα + FLK1− cells from Ldb1 knockdown experiments were cultured 4 days as monolayer (equivalent to 6-day Pax3 induction) and collected for ChIP, HiChIP and gene expression analysis or withdrawn of bFGF and dox for 2 days and collected for immunostaining and western blot.

**Ethics statement**. All animals were handled in strict accordance with good animal practice as defined by the relevant national and/or local animal welfare bodies, and all animal work was approved by the University of Minnesota Institutional Animal Care and Use Committee (protocol number 1702-34580A).

**Mice and embryo explants**. Ldb1 deletion in the Pax3 lineage was achieved by timely mating Pax3cre/+; Ldb1fl/+ males with Ldb1fl/fl females. Importantly, for these experiments the Cre was inherited through the paternal lineage. For immunostaining, embryos were collected at 9.5, 10.5, or 11.5 d.p.c., embedded in Tissue-Tek® O.C.T. compound (Sakura® Finetek—VWR) and frozen. For all experiments, genotyping was performed using genomic DNA isolated from Yolk Sacs.

**Chromatin-immunoprecipitation**. ChIP was performed following the protocol described by Young and colleagues with minor modifications[64]. Briefly, d4 EBs (equivalent to non-induced and 1-day induced) were trypsin-treated at 37 °C for 1 min with gentle shaking and reaction was inhibited by adding 10%FBS/PBS. Single cells were washed once with PBS, resuspended in 10%FBS/PBS and supplemented with formaldehyde (final concentration 1%) for crosslinking of protein-DNA complexes (10 min at RT) followed by quenching with glycine and staining with PDGFRα-PE antibody. PDGFRα+ cells were sorted using FACSAriaII, snap-frozen in liquid nitrogen and stored at −80 °C if not processed immediately. For 6-day experiments, PDGFRα+FLK1− cells were FACS sorted from d5 EBs (equivalent to 2-day induction) and then cultured for additional 4 days (as described in the Cell culture section) before collecting them for formaldehyde crosslinking. Cell pellets were incubated in lysis buffer LB1 supplemented with protease inhibitors (50 mM HEPES KOH pH 7.5, 140 mM NaCl, 1 mM EDTA, 10% glycerol, 0.5% NP40, 0.25% Triton X-100 + Complete-mini—Roche) for 10 min at +4 °C followed by

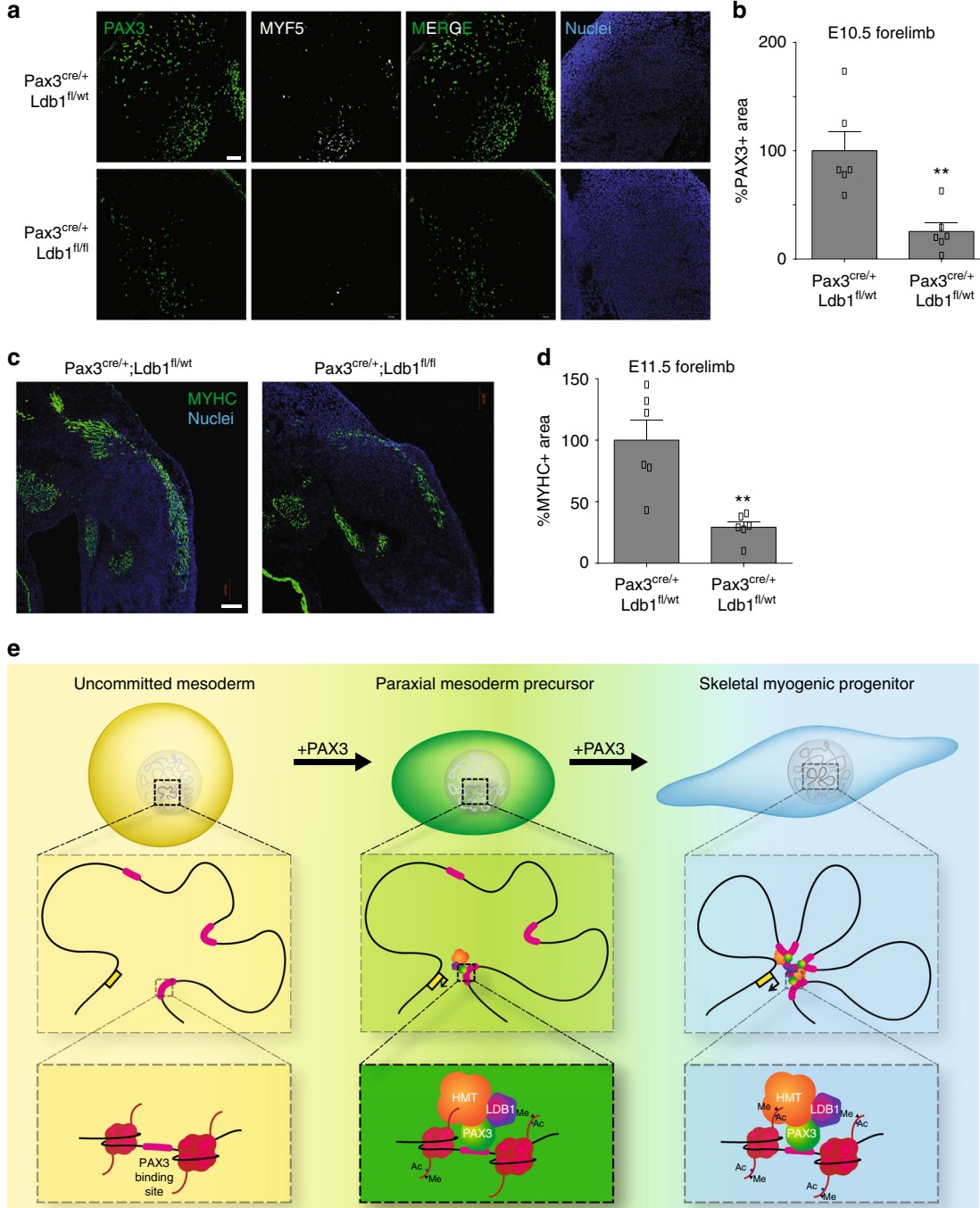

**Fig. 7** Ldb1 is required for hypaxial myogenesis. **a** Myogenic progenitor migration is impaired upon Ldb1 deletion in the Pax3 lineage. Representative immunostaining of cryosections from E10.5 Ldb1-deleted embryos reveals reduced numbers of PAX3+ cells (green) in the forelimb bud. MYF5 (white); nuclei (blue). Bar: 100 μm. Images are representative of at least three independent embryos (from distinct matings). **b** Quantification of PAX3 + signal from panel **a**. Graph represents mean + s.e.m. from three independent embryos ($n = 3$) (D). Student's $t$-test *$p < 0.05$. **c** Immunostaining of E11.5 cryosections using MYHC antibody confirms the impaired migration observed in E10.5 embryos. MYHC (green); nuclei (blue). Bar: 100 μm. **d** Quantification of MYHC + signal from panel **c**. Graph represents mean + s.e.m. from independent embryos ($n = 3$). Student's $t$-test ***$p < 0.001$. **e** Schematized model of Pax3-induced myogenic commitment. Pax3 regulates chromatin remodeling and looping by recruiting the histone methyltransferase (HMT) complex and Ldb1. Source data are provided as a Source Data file

incubation in buffer LB2 supplemented with protease inhibitors (10 mM Tris-HCl pH 8, 200 mM NaCl, 1 mM EDTA, 0.5 mM EGTA + Complete-mini—Roche) for 10 min at +4 °C. Cell pellet was then resuspended in LB3 supplemented with protease inhibitors (10 mM Tris-HCl pH 8, 100 mM NaCl, 1 mM EDTA, 0.5 mM EGTA, 0.1% sodium deoxycholate, 0.5% N-lauroylsarcosine + Complete-mini—Roche) and then sonicated. For Histone marks, cells were sonicated using Bioruptor Pico (25 cycles 30″ ON, 30″ OFF) to achieve an average size of 200 bp. For transcription factors and cofactors, cells were sonicated with a Branson sonicator at 18% power for 1 min with intervals of 10 s ON–10 s OFF to achieve an average chromatin size of 300 bp. After shearing, samples were centrifuged for 10 min at 16,000 × $g$ and snap-frozen in liquid nitrogen if not processed immediately. For Histone ChIP, 12 μg of chromatin (diluted to 250 μl) were precleared for 4 h at 4 °C with 10 μl of BSA-blocked Protein A-conjugated sepharose beads (GE Healthcare). For TF-cofactor ChIP, 25–40 μg of chromatin (diluted to 500 μl) were precleared for 4 h at 4 °C with 20 μl of BSA-blocked Protein A (or Protein G)-conjugated sepharose beads (GE Healthcare). Samples were supplemented with 1/10 volume of

10% Triton X-100 and incubated overnight with antibodies for immunoprecipitation. The list of antibodies used in this study and relative dilutions are provided in Supplementary Table 2. Immune complexes were recovered by incubation with 20 µl of BSA-blocked Protein A (or Protein G)-conjugated sepharose beads for 4 h at 4 °C and then washed five times with RIPA wash buffer (50 mM HEPES KOH pH 7.5, 500 mM LiCl, 1 mM EDTA, 1% NP40, 0.25% Triton X-100, 0.7% sodium deoxycholate) and one time with TEN buffer (10 mM Tris-HCl pH 8, 1 mM EDTA, 50 mM NaCl). Immunoprecipitated chromatin was recovered by incubating beads with 200 µl of Elution buffer (50 mM Tris-HCl pH 8, 10 mM EDTA, 1% sodium dodecyl sulfate) for 20 min at 65 °C. Chromatin from IP and Input (equivalent to 1% of starting material) was reverse crosslinked overnight at 65 °C, then diluted 1:1 with TE (10 mM Tris-HCl pH 8, 1 mM EDTA) supplemented with 4 µl of RNaseA 20 mg/ml and incubated for 2 h at 37 °C followed by Proteinase K treatment (4 µl of 20 mg/ml stock for each sample) for 30 min at 55 °C. DNA was purified by phenol–chloroform–isoamyl alcol extraction (twice) followed by chloroform extraction, then supplemented with 1/10 of volume of 3 M sodium acetate pH 5 and 1.5 µl of glycogen and precipitated with two volumes of 100% ethanol at 180 °C for >1 h. Followed 30 min centrifuge at $16,000 \times g$, pellet was washed with 75% ethanol, air dried and dissolved in 45 µl $H_2O$. qPCR was performed in a final volume of 10 µl using SYBR Premix Ex Taq II (Clontech), 0.5 µl of 1.4 µM primer stock, and 0.3 µl sample/reaction and run on a 384-well plate on a ABI7900HT instrument (Applied Biosystems). The primer sequences used for qPCR are provided in Supplementary Table 1.

**ChIP-seq library generation.** Libraries were generated following a gel-free protocol using AMPure XP beads (Beckman Coulter) for all the purification and size selection steps. Ten nanograms or less of DNA were end repaired using End-it DNA end repair (Epicentre), then A-tailed using Klenow Fragment ($3' \rightarrow 5'$ exo-NEB) followed by adapter-barcode ligation using T4 DNA ligase (Enzymatics). Illumina compatible adapter-barcodes were purchased from BIOO scientific. After ligation, DNAs were negatively size selected using 0.5× Ampure XP beads and unbound DNAs were positively size selected by adding 0.4× Ampure XP beads (this step allows for retention of DNA fragments ranging 200–500 bp). Libraries were amplified using Phusion High Fidelity PCR master mix 2x (NEB) with a 16 cycles program. Ldb1 and H3K4me1 (from ΔTAD and ΔTAD-Ldb1 lines) ChIP-seq libraries were generated using the NEBNext UltraII DNA library prep kit (NEB) following manufacturer's instructions. Purified libraries were then submitted to the University of Minnesota Genomic Center (UMGC) for quantification, quality control, and sequencing. Libraries were pooled and sequenced on Single-End runs of the HiSeq2500 operated at High Output mode (Illumina). Libraries for TF/cofactors were sequenced to an average depth of 25 million reads/sample. Libraries for histone marks were sequenced to an average depth of 45 million reads/sample (Biological replicate 1) and 35 million reads/sample (Biological replicate 2). The sequencer outputs were processed using the computer cluster managed by the Minnesota Supercomputing Institute (MSI).

**ChIP-seq analysis.** Each sample's reads were aligned to the mouse genome (mm10) using Bowtie2[65] followed by removal of PCR duplicates using Samtools[66]. Peak calling was performed using MACS[67] with the following parameters: –bw 300 -p 1e-3. Similar results were obtained by performing peak calling using QESEQ[68] with the following parameters: transcription factors/cofactors -s 100 -c 15 -p 0.01; histones marks -s 100 -c 20 -p 0.001 (replicate 1) -s 100 -c 15 -p 0.001 (replicate 2). To identify the list of high confidence PAX3 and LDB1 peaks we performed three independent ChIP-seq experiments and, using the MACS output and the bedtools intersect function, only common regions between two experiments were further considered[69]. The Myf5 −111 kb region was detected in only one of three biological replicates generated for 1-day PAX3 and 1-day LDB1 ChIP. However, upon validation by qPCR (Fig. 6c and Supplementary Fig. 1e), this region was included in the list of 1-day PAX3 and LDB1 peaks. SMC1 and CTCF peaks were defined as the common peaks among two independent ChIP-seq datasets (using bedtools intersect). In addition, peak lists were filtered (intersect –v option) for sites overlapping to peaks detected in the uninduced control ChIP-seq and in the mouse ChIP-seq black-list (Consortium, 2012). Bigwig files for visualization on IGV[70] were generated by converting the wig files obtained from MACS. k-means clustering maps were generated with SeqMiner using the enrichment file enr.sgr produced by QESEQ. Gene annotation of Pax3 peaks was performed using PAVIS (https://manticore.niehs.nih.gov/pavis2/)[71]. Functional annotation of ChIP-seq clusters was performed using GREAT (http://bejerano.stanford.edu/great/public/html/)[41].

**VISTA enhancers.** Genomic coordinates and embryo pictures for the enhancers annotated to Somite, Limb, and Neural tube were downloaded from the Vista browser (https://enhancer.lbl.gov/)[27]. Identification of the Pax3 peaks overlapping to Vista enhancers was performed using bedtools intersect. Resulting bed files were loaded on IGV for visualization.

**HiChIP.** Long-range interactions involving Pax3 binding sites were analyzed following the detailed protocol described by the Chang group[24]. Briefly, non-induced, 1-day and 6-day Pax3-induced cells (mesoderm, paraxial mesoderm precursors,

and myogenic progenitors, respectively), 6-day + shSCR and 6-day + shLdb1 and non-induced 1-day ΔTAD-Ldb1-induced cells were harvested using trypsin for 1–2 min, followed by inactivation with PBS/10% FBS, centrifuged for 5 min at $400 \times g$, washed with PBS twice, and then counted. Thirteen million cells, diluted to 1 million/ml in PBS, were crosslinked for 15 min using methanol-free formaldehyde (Pierce—Thermoscientific). In situ contact generation was performed by digesting chromatin with MboI (NEB) followed by blunting with dNTP mix containing biotin-dATP (Thermo Fisher, 19524016) and ligation using T4 DNA ligase (NEB). After centrifugation, nuclei were resuspended in 880 µL in Nuclear Lysis Buffer (50 mM Tris-HCl pH 7.5, 10 mM EDTA, 1% SDS, Complete Protease inhibitor—Roche), transferred in a Millitube and sonicated using a Covaris S220 (parameters: fill level = 10, duty cycle = 5, PIP = 140, cycles/burst = 200, time = 4 min). Chromatin was cleared by centrifugation, diluted 1:2 in ChIP Dilution Buffer (0.01% SDS, 1.1% Triton X-100, 1.2 mM EDTA, 16.7 mM Tris-HCl pH 7.5, 167 mM NaCl), precleared for 4 h at +4 °C with 75 µl of Protein G-conjugated magnetic beads and then incubated with 30 µg of anti-Pax3 antibody overnight at +4 °C. For ΔTAD-Ldb1 HiChIP, chromatin was precleared with Protein A beads and then incubated with 15 µg of anti-Ldb1 antibody (Abcam) overnight at +4 °C. Immune complexes were recovered by incubation with 60 µl of BSA-blocked Protein G (or Protein A for Ldb1 HiChIP)-conjugated sepharose beads for 5 h at +4 °C, then washed five times with RIPA wash buffer (50 mM HEPES KOH pH 7.5, 500 mM LiCl, 1 mM EDTA, 1% NP40, 0.25% Triton X-100, 0.7% sodium deoxycholate) and one time with TEN buffer (10 mM Tris-HCl pH 8, 1 mM EDTA, 50 mM NaCl). Due to the large sample volume during Pax3/Ldb1 HiChIP, Protein-A/G:Antibody-chromatin bound beads were splitted in two 1.5 ml tubes during the previous procedure. Immunoprecipitated chromatin was recovered by incubating beads with 200 µl/tube of Elution buffer (50 mM Tris-HCl pH 8, 10 mM EDTA, 1% sodium dodecyl sulfate) for 30 min at 65 °C. ChIP samples and Input (equivalent to 1% of starting material) were reverse crosslinked overnight at 65 °C, then diluted 1:1 with TE (10 mM Tris-HCl pH 8, 1 mM EDTA) supplemented with 4 µl of RNaseA 20 mg/ml and incubated for 2 h at 37 °C followed by Proteinase K treatment (4 µl of 20 mg/ml stock for each sample) for 30 min at 55 °C. DNA was purified by phenol–chloroform–isoamyl alcol extraction (twice) followed by chloroform extraction, then supplemented with 1/10 of volume of 3 M Sodium Acetate pH 5 and 1.5 µl of glycogen and precipitated with two volumes of 100% ethanol at −80 °C for >1 h. Following 30 min centrifuge at $16,000 \times g$, pellets were washed with 75% ethanol, air dried and the combined DNA pellet were dissolved in 40 µl $H_2O$. Before proceeding with the biotin capture and transposase mediated library generation, ChIP samples were analyzed by qPCR to ensure enrichment of Pax3 sites (Myf5 −111 kb, Vcam1 +3 kb) vs. control region (Myf5 −257 kb). Samples were quantified using Pico-green (Invitrogen), resuspended in binding buffer and then incubated with 20 µl of Streptavidin T1-conjugated magnetic beads (Invitrogen). After washing, DNA-bound streptavidin beads were incubated with 1.5 µl of Tn5 transposase in a final volume of 50 µl for 10 min at 55 °C. This amount of transposase was selected based on the picogreen results of the ChIP, which yielded 25 ng of DNA. Following quenching and washing, transposed DNA was incubated 5 min at 65 °C with activated NEBNext UltraII QS Taq (+Illumina adaptor/barcodes) to allow primer extension and then amplified for five cycles (15 s 98 °C followed by 1.5 min at 65 °C). Additional cycles for library amplification were determined by qPCR using T5-T7 primers, then libraries were purified with 1.5× AMPure beads and eluted in 50 µl of $H_2O$. After Quality Control, libraries were pooled and sequenced on a Paired-End run of the HiSeq2500 operated at High Output mode (Illumina) which yielded ~80 million reads for each of the two biological replicates.

**HiChIP analysis.** HiChIP paired-end samples were processed using the HiC-Pro pipeline[72]. Contact matrices were generated for 10-kb bin pairs using default parameters in HiC-Pro to map the reads to the mm10 genome, remove duplicate reads, assign reads to MboI restriction fragments, and acquire valid interactions. Statistically significant bin pairs and number of contacts were determined using the Fit-Hi-C contact caller using default settings and interaction distance <2 Mb[73].

**Generation of contact matrix heat maps.** Raw contact matrices from the HiC-Pro pipeline were normalized by read depth. The resulting normalized matrices were used to generate heat maps. The matrix acquired for the condition with no addition of doxycycline (non-induced) was subtracted from the matrices for 1-day and 6-day in doxycycline conditions. This enhanced the visualization of the heat maps for contacts specific to the manipulated condition. Similarly, the matrix for the Ldb1 knockdown sample was subtracted from the shSCR matrix to emphasize loss of contacts with the knockdown of Ldb1. R Bioconductor package "HiTC" was used to assist the generation of the contact heat maps[74].

**Determination of significant interactions.** FitHiChIP was used for the determination of statistically significant interactions[75]. Initially, peaks were called from the aligned reads generated by the HiC-Pro pipeline. Using these called peaks along with the default parameters of the FitHiChIP algorithm, significant interactions were collected with a stringency q-value cutoff of 0.01. The interactions from the control conditions were subtracted (bedtools intersect –v –f 0.9 –F 0.9) to filter for interactions specific only to the condition of interest. The resulting interaction

coordinates for both anchors of each loop were overlapped with ChIP-seq peaks to quantitatively compare the HiChIP results with ChIP-seq analysis. Visualizations of these significant loops were viewed on WashU Epigenome Browser[76].

**Western blot and immunoprecipitation**. Proteins were extracted from cultured cells using RIPA buffer (150 mM NaCl, 50 mM Tris-HCl pH 7.5, 1 mM EDTA, 1% Triton, 1% sodium deoxicholate, 0.1% SDS) supplemented of Protease inhibitors (Complete—Roche) and quantified with Bradford reagent (Sigma-Aldrich). Protein samples were prepared in Laemmli buffer and loaded on gels for SDS-PAGE. Proteins were transferred on PVDF membranes (Millipore) for the detection with the indicated antibodies. The list of antibodies used in this study and relative dilutions are provided in Supplementary Table 2. Isolation of the Pax3 complex was performed using ~300 million cells from Pax3- and Pax3HaFlag-induced mES cells. Doxycycline was added on day 3 of differentiation at the final concentration of 2 μg/ml. Day-4 EBs (equivalent to non-induced and 1-day induced cells) were disgregated using trypsin and gentle shaking in a 37 °C water bath followed by inactivation with FBS and gentle resuspension with pipette. After centrifuge (5 min at 500 × g), cell pellets were washed with ice cold PBS and incubated 30 min in 10 packed cell volumes (PCV) of ice cold PBS supplemented with 5 μg/ml Digitonin (Sigma-Aldrich). Cells were centrifuged 5 min at 500 × g, gently resuspended in five PCV of ice cold Hypotonic buffer (20 mM Tris-HCl pH 7.3, 1.5 mM MgCl$_2$, 10 mM KCl, 1.5 mM DTT supplemented with PMSF, protease (Complete—Roche), and phosphatase (PhoStop—Roche) inhibitors), incubated 10 min on ice and vortexed for 10 s. After centrifuge (10 min at 500 × g), the previous step was repeated once and the nuclei preparation was checked on a microscope using Trypan blue. Nuclei were then gently resuspended in one PCV of ice cold low salt buffer (10 mM Tris-HCl pH 7.3, 100 mM NaCl, 1.5 mM MgCl$_2$ + PMSF, protease and phosphatase inhibitors) and then supplemented drop-by-drop with one PCV of ice cold High Salt Buffer (50 mM Tris-HCl pH 7.3, 800 mM NaCl, 1.5 mM MgCl$_2$, 25% glycerol + PMSF, protease and phosphatase inhibitors) and 4 μl of Benzonase (Sigma-Aldrich). After 30 min incubation on ice, sample were centrifuged 30 min at +4 °C 16,000 × g and the supernatant was transferred into 3 ml Slide-A-Lyzer® G2 Dialysis Cassette—cutoff 7000 MWCO (Thermoscientific) and dialyzed overnight at +4 °C with gentle stirring using Dialysis buffer (10 mM Tris-HCl pH 7.3, 100 mM NaCl, 1.5 mM MgCl$_2$, 0.1 mM EDTA, 10% glycerol). Nuclear extracts were recovered, centrifuged 30 min at +4 °C 16,000 × g to remove precipitated proteins and precleared 2 h with 300 μl of Agarose IgG beads. This method yielded ~10 mg of nuclear extract, which was then incubated with 200 μl of anti-FLAG M2 magnetic beads (Sigma-Aldrich) for 4–5 h at +4 °C with gentle rotation. Aliquots were collected before (Input), after (FT) FLAG IP for quality control. Beads were washed five times with 10 volumes of Wash buffer_1 (10 mM Tris-HCl pH 7.3, 100 mM NaCl, 1.5 mM MgCl$_2$, 0.1 mM EDTA, 10% glycerol, 0.1% NP-40) and then incubated for 1 h with 150 μl of 0.2 mg/ml 3xFLAG peptide diluted in Wash buffer_1. This step was repeated four more times and, ultimately, all elutions were pulled and incubated with 100 μl of BSA-blocked anti-HA magnetic beads (Pierce—Thermoscientific) for 5–6 h at +4 °C with gentle rotation. Aliquots were collected before (termed IP αFLAG) and after (FT 2nd IP) HA immunoprecipitation for quality control. Beads were washed five times with 10 volumes of Wash buffer_2 (10 mM Tris-HCl pH 7.3, 200 mM NaCl, 1.5 mM MgCl$_2$, 0.1 mM EDTA, 10% glycerol, 0.1% NP-40) and then incubated for 1 h with 100 μl of 0.2 mg/ml HA peptide diluted in Wash buffer_1. This step was repeated three more times with 50 μl of 0.2 mg/ml HA peptide diluted in Wash buffer_1 and, ultimately, all elutions were pulled. An aliquot of the final elution (IP αHA) was collected for quality control. Aliquots from the immunoprecipitation were analyzed by western blot using a monoclonal Pax3 antibody (DHSB), and silver staining using a 4–15% gradient polyacrylamide gel (Biorad). Mass spectrometry analysis was performed at the Taplin Mass Spectrometry facility (Harvard Medical School). Two biological replicates were submitted as protein pellets following trichloroacetic acid (TCA) precipitation. However, due to the lower yield from the control sample (IP from untagged Pax3 line), these two replicates were combined before mass spectrometry analysis to ensure the detection of non-specific interactors.

**Mass spectrometry sample preparation**. Pellets were resuspended with ammonium bicarbonates and reduced by adding DTT at a 1 mM concentration (in 50 mM ammonium bicarbonate) for 30 min at 60 °C, then cooled to room temperature and supplemented with iodoacetamide (stock in 50 mM ammonium bicarbonate) to a concentration of 5 mM followed by incubation for 15 min in the dark at room temperature. DTT was then added to a 5 mM concentration to quench the reaction. Digestion was performed by adding sequence grade trypsin at a concentration of 5 ng/μl and overnight incubation at 37 °C. Samples were then desalted by an in-house made desalting column.

**Mass spectrometry run**. On the day of analysis, samples were reconstituted in 5–10 μl of HPLC solvent A (97.5% water, 2.5% acetonitrile, 0.1% formic acid). A nano-scale reverse-phase HPLC capillary column was created by packing 2.6 μm C18 spherical silica beads into a fused silica capillary (100 μm inner diameter × ~30 cm length) with a flame-drawn tip[77]. After equilibrating the column each sample was loaded via a Famos auto sampler (LC Packings, San Francisco, CA)

onto the column. A gradient was formed and peptides were eluted with increasing concentrations of solvent B (97.5% acetonitrile, 2.5% water and 0.1% formic acid). As peptides eluted, they were subjected to electrospray ionization and then entered into an LTQ Orbitrap Velos Pro ion-trap mass spectrometer (Thermo Fisher Scientific, Waltham, MA). Peptides were detected, isolated, and fragmented to produce a tandem mass spectrum of specific fragment ions for each peptide.

**Data processing protocol**. Peptide sequences (and hence protein identity) were determined by matching protein databases (UniProt) with the acquired fragmentation pattern by the software program, Sequest (Thermo Fisher Scientific, Waltham, MA)[78]. All databases include a reversed version of all the sequences and the data was filtered to between 1–2% peptide false discovery rate (FDR). Search settings: peptide_mass_tolerance = 2.0; digest_mass_range = 600.0000 35000.0000; max_num_internal_cleavage_sites = 2; diff_search_options = 15.9949146221 M 14.0157 C; add_C_Cysteine = 57.0214637236. The list of proteins detected by mass spectrometry (available in Supplementary Data 4) was filtered to exclude candidates equally detected in control and IP sample. For the initial analysis, we also applied a cutoff of five unique peptides per protein in order to identify high confidence candidates. The filtered list was analyzed using STRING (available at www.expasy.org), which enabled the identification of protein complexes associated to Pax3. List of proteins used for STRING analysis and the results are available in Supplementary Data 5. Validation of Mass spectrometry candidates was performed by WB using samples from FLAG-IP. In addition, immunoprecipitation using the rabbit anti-LDB1 antibody was performed on nuclear extracts from iPax3 and iPax7 lines followed by pull-down of the immune complexes with sepharose-conjugated Protein A beads. List of antibody used in this study is provided in Supplementary Table 2.

**RNA isolation, qRT-PCR, and transcriptomic analysis**. Samples were resuspended in Trizol (Invitrogen) prior to RNA isolation using the PureLinkTM RNA Mini kit (Invitrogen) following the manufacturer's instructions for "Trizol samples" (including in-column DNase treatment). RNAs were retro-transcribed using Superscript Vilo (Invitrogen). Gene expression analyses were performed using an amount of cDNA corresponding to 12.5 ng of starting RNA for each reaction. qRT-PCR were performed using Premix Ex Taq (Probe qPCR) Master Mix (Takara) and TaqMan probes (Applied Biosystems).

The 1-day/6-day Pax3 induction transcriptomic analysis was previously described[20]. For heatmap in Supplementary Fig. 6c, differentially expressed genes were determined by annotation of each ChIP-seq peak to the closest gene. The resulting list was then filtered based t test analysis (p-value < 0.05) and read count >10 and fold induction/repression >2. Heatmap was generated using the R package pheatmap.

**Immunostaining**. Immunofluorescence staining was performed by fixing cells with 4% Paraformaldehyde/PBS for 10 min at 4 °C, then permeabilized with 0.1% Triton/PBS and blocked with 5% BSA/PBS before incubating with the primary antibodies. The list of antibodies used in this study and relative dilutions are provided in Supplementary Table 2. Samples were rinsed with PBS, blocked with 5% BSA/PBS and then incubated with the secondary antibody. After washing, samples were mounted on the slides using Prolong Gold with DAPI (Invitrogen). Embryo cryosections (12-μm thick) were washed with PBS + 0.1%Tween (PBST), incubated for 1 h at RT in Permeabilization-Blocking solution (5% Normal Donkey Serum (Jackson Immunoresearch Laboratories), 0.1% Triton X-100 in 1× PBS) and then overnight at +4 °C with primary antibodies diluted in Antibody Diluent (Dako). Slides were washed three times with PBST, incubated with secondary antibodies diluted in Antibody Diluent (Dako) for 1 h at RT, washed again three times with PBST and briefly dried before mounting using Prolong Gold with DAPI (Invitrogen). Pictures were acquired with Axioimager M1 fluorescence microscope or LSM 510 Meta confocal microscope (Zeiss).

**Quantification and statistical analysis**. Analyses were performed using the ImageJ distribution Fiji[79]. Analysis of myogenic progenitor migration and differentiation in E10.5 and E11.5 forelimbs was performed as follow: color channels were separated and threshold level for the green and blue channels were adjusted in order to select the area positive respectively to PAX3 or MYHC (green) and DAPI (blue). The area positive for each channel was analyzed using Analyze Particle using 0-Infinity as Size parameter. Finally, the value representing PAX3+ and MYHC+ area for each image was normalized based on nuclear staining (DAPI+). After calculating mean, all values were reported as % relative to mean for graphical display. Data represent mean ± s.e.m. of two representative pictures from three independent experiments. Statistical analyses between control and treated group in Fig. 7b, d were performed using unpaired two-tailed Student's t-test. P values < 0.05 were considered to be statistically significant.

**Reporting summary**. Further information on research design is available in the Nature Research Reporting Summary linked to this article.

## Data availability

Sequencing data from this work are available under the GEO accession numbers GSE126362 and GSE125203. The mass spectrometry proteomics data have been deposited to the ProteomeXchange Consortium via the PRIDE[80] partner repository with the dataset identifier PXD012693. All other relevant data supporting the key findings of this study are available within the article and its Supplementary Information files or from the corresponding authors upon reasonable request. The source data underlying Figs. 3b, c, e, 5a, c, d, 6c, e and 7b, d and Supplementary Figs. 1b, d, e, 2b, c, 6a, c, d, 7b and 8a-e are provided as a Source Data file. A reporting summary for this Article is available as a Supplementary Information file.

## Code availability

All data analyses were performed using publicly available software.

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

## Acknowledgements

This project was supported by NIH grants R01 AR055299 and AR071439 (R.C.R.P.), U01 HL100407R01 (R.C.R.P. and D.J.G.), R21 AR068786 (B.D.D.), R01 GM122395 (B.D.D.), and by Regenerative Medicine Minnesota (A.M.). C.O.C. was supported by PINN MICITT Costa Rica. The authors thank the continuous support of Dr. James Thomson, Dr. Ron Stewart and Scott Swanson from the Morgridge Institute for Research (Madison, WI). We also thank the help from: Dr. Gabriel Starrett (computational analyses); Yi Ren (Lillehei Heart Institute FACS core); the staff of the University of Minnesota Genomic Center (technical support); Bridget Dillon, Asma Redwan, and Bayardo Garay (technical assistance). The authors acknowledge the Minnesota Supercomputing Institute (MSI) at the University of Minnesota for providing resources that contributed to the research results reported within this paper. URL: http://www.msi.umn.edu. In addition, the authors would like to thank Dr. Michael Kyba and Dr. Daryl Gohl for critical reading of the manuscript and Cynthia Faraday for assistance with artwork.

## Author contributions

A.M. designed and performed the research, performed bioinformatic analyses, analyzed the data and wrote the manuscript; J.B. and C.O.C. performed the research and analyzed the data; P.P. and I.Y.K. performed bioinformatic analyses; D.J.G. supervised the bioinformatic analyses; P.E.L. and B.D.D. supervised research; R.C.R.P. contributed with experimental design, interpretation of the data and wrote the manuscript.

## Additional information

**Competing interests:** The authors declare no competing interests.

