## [Peer Review File · Nature Communications]

Reviewers' Comments:

Reviewer #1:

Remarks to the Author:

Magli et al.

Pax3 reprograms local chromosome architecture during lineage specification

This study investigates the role of transcription factors in remodelling chromatin during commitment of myogenic cells. Using HiChIP and mass spectrometry assays of Pax3 interacting partners, the authors identify the widely expressed Ldb1 transcription factor as a critical component in Pax3-mediated chromatin looping during myogenic commitment in an in vitro differentiation model. They subsequently perform analysis and specific gene KO experiments in vivo to demonstrate that Ldb1 is critical for maintaining the number of Pax3+ cells in the developing mouse limb bud. The results and conclusions of the study provide a significant advancement of our understanding of how chromatin is regulated in multiple steps via an interplay between general regulators (Ctcf, cohesion) and transcription factors during myogenic cell commitment. There are a few concerns that the authors need to address to clarify and improve the study. These are listed below.

One point concerns the title – the word "reprograms" might be out of context here. Perhaps "modifies" or "directs", or some similar word would be more appropriate. Also, some functional data is provided for Ldb1, so indicating this TF in the title is suggested.

Comments:

- 1) Some more information on the protocol for induction and differentiation is needed to understand the homogeneity of the cells being analysed and the differentiation scheme. In the methods, Pax3 was induced in 3day EBs and analysis was done at Days 1 and 6. The scheme indicates ES and mesoderm precursors. It would be more accurate to indicate EBs instead of "mesoderm precursor". Also, the authors induce differentiation using a Dox system of Pax3 expression. Please show staining of the cells at Day 1 and Day 6 (assay points; also pre-assay points as controls) for Pax3, Myf5, MyoD and Myogenin staining. The detection of Myf5 transcripts at Day 1 is not a sufficient indicator of what is happening at the single cell level and the dynamics of the state of the cells using this protocol. The authors need to indicate these percentage values for the assay times used.
- 2) Please explain why it is expected that Pax3 occupies -111 and -57 positions in the Myf5 locus. It is not clear what these elements are doing, what they represent, and if Pax3 binds these sites in the embryo.
- 3) The Rudnicki lab reported genome-wide analysis of Pax3 and Pax7 in adult myogenic cells. It would be informative to compare that data with the one presented in this study.
- 4) Page 4: "peaks are associated with genes located many kilobases away". Do the authors mean sequences or genes? Away from what landmark?
- 5) Page 4: the authors give as examples "Gas1 +29kb, Megf10 +135kb, Meis2 +144kb, Myf5 -111kb, MyoD -27kb and Vcam1 +3kb" for long range interactions. It is not clear what exactly constitutes long-range, but "+3kb" might not fit into this definition. The authors need to clarify for the general reader how these distinctions are made.
- 6) Figure 1B: there appears to be a strong peak and a couple of smaller peaks for Pax3 in the Myf5 gene at Day 1 but not at Day 6. Please comment on the significance of this, as it might be expected to be the other way around.
- 7) Figure 2A: please indicate why Chr12 was chosen? Are Pax3 targets more enriched on this

chromosome? Chr10 contains Myf5, would this have been a better choice for analysis? Also in Figure 2B, E, G, please explain the significance of the different Histone marks marking the Pax3+ enhancers.

8) Please explain the significance of the Histone modifications for the general reader and their relationships amongst each other, as well as annotations of Figures (ex. Red box = 9 Figure 2B). For example, Fig S2A, there are correlative and anti-correlative (Cluster 2 and 3) peaks for K4me1 and K27ac in the examples given – why is this the case if both should mark enhancers.

9) Please indicate in all Figures the source of the cells and use uniform timing scheme throughout. Ex. Fig 4B indicates 4 day EBs – which correlates to Day 1. What is the timing for the other experiments, and their n-values. A supplementary Figure on replicates is provided for some experiments, but this is not obvious for all of the experiments in the study. All the n-values need to be indicated in the legends. Similarly, how was the data calculated in Figure 5 for Pax3 and MYHC stainings? Although 3 independent embryos were analysed, how many sections were used, and how was the area calculated? This is a rather imprecise quantification method for Pax3 given the nuclear staining, however it might be ok for MYHC quantification.

10) The example provided in Figure 5C is too faint to assess what is happening to Myf5 staining. Also, Fig S4B, the "nuclei" panel does not contain only nuclei.

11) Indicate more clearly in Figure legends for 4D and F what is the difference between category 1 and 2.

12) Although the authors focused their study on the embryo, Pax3 is largely downregulated in the mouse fetus. It would be interesting to determine if the same principles proposed for gene regulation apply to Pax7. It is not necessary to perform a detailed study as has been done, but nonetheless, it would be highly informative and complementary to provide a wider message to readers, as a paradigm-principle for the model that the authors propose, and that is not restricted only to analysis of embryonic cells.

Reviewer #2:

Remarks to the Author:

The manuscript submitted by Magli et al is interesting and contains novel information about Pax3 and Ldb1 but is premature for publication in its current form. Perhaps the most notable problems are that a main conclusion is not adequately supported and the genome-wide data sets are not adequately interrogated or presented. In addition, it appears that the different assays were performed at different times, raising questions about the validity of the data comparisons.

1) A major conclusion of the work is that the Ldb1 protein is "the mediator of looping interactions at Pax3 binding sites" (quoted from abstract). The Ldb1 depletion studies nicely show biological relevance in myogenesis, but the connection to looping is limited to decreased deposition of H3K4me1 at sites occupied by Ldb1 and Pax3. Such a result simply does not show Ldb1 mediates looping, and is it important to note that even though the data (Fig. 5B) are statistically significant, the decrease in deposition of H3K4me1 at the sites is less than 2-fold in every instance. It is hard to understand how a very modest decrease in this histone modification leads to the aforementioned conclusion. The clearest experiment to address the issue is to compare the Hi-ChIP results in control and Ldb1 knockdown cells.

2) The authors also correlate loop formation between Pax3 bound sequences with gene expression, but the study lacks direct evidence that loop formation is required for transcription.

3) The authors present multiple genome-wide studies but fail to present the results, preferring to pick out specific loci or other specific information and solely present that.

a) Pax3 ChIP-seq: No information is given on the number of reads and number of mapped reads. The number of identified peaks is not given. Was it 20 or 20,000? Could a supplemental table be presented identifying the binding sites? Are there distinguishing differences in Pax3 binding between day 1 and day 6 samples, and if so, what are they? What is the relationship between the location of Pax3 binding sites and genes? Importantly, the authors conclude that most Pax3 binding sites are distant from genes. What determines whether a binding site is distant from a nearby gene vs a binding site being intergenic? Perhaps comparison of Pax3 binding sites to regions of RNA pol II binding might better inform the relationship between Pax3 localization and gene expression.

b) The same concerns and questions apply to the Ldb1 and Smc1 ChIP-seq data. It is simply not appropriate to do a genome wide study and then provide no data or analysis whatsoever except for the very small subset of the data that relates to Pax3 binding.

c) Hi-ChIP: No information is given about the number of interactions observed across the genome and whether the number of observed interactions is evenly distributed amongst Pax3 binding sites or whether some Pax3 binding sites are "hotspots" for interactions. Is there bias in the frequency of interactions based on chromosome (ie – are interactions more frequent on specific chromosomes or on "gene-rich" chromosomes)? Is there a way global interactions could be summarized in tabular or graphical form? Would it be too much to confirm a few of the selected interactions by conventional 3C?

4) There are disconnects in the data comparisons. It appears the Pax3 ChIP-seq and the Hi-ChIP are comparisons of day 1 and day 6 post induction of Pax3. It seems all of the other ChIP-seq analyses of histone modifications are from day 4 embryoid bodies. Does this correspond to 4 days post-induction of Pax3? The legend to Figure 2C suggests that this is the case. So why is it valid to compare histone ChIP-seq from day 4 to Pax3 ChIP-seq and Hi-ChIP on days 1 and 6? In Figure S3B, the authors present RNA-seq from day 5. How does this relate to the other studies done on different days?

In addition, the authors consistently use the phrase "chromatin remodeling" when talking about the chromatin landscape defined by the histone modifications that were assayed. Many (most?) people use "chromatin remodeling" to talk about ATP-dependent enzymatic alterations to chromatin and use chromatin or histone "modifications" to talk about the various chemical groups that can be added and removed from the histones. The authors should consider this point and justify their use of "chromatin remodeling" or change the text throughout.

5) In Figure 1D, the authors state that long-range interactions are clearly defined for specified genomic loci. However, it is not possible to associate the regions indicated with the density matrices. The authors should consider alternate presentations of the interactions, possibly in the manner of data presented in other Hi-ChIP papers such as Mumbach et al (<https://www.ncbi.nlm.nih.gov/pubmed/27643841>) or Nora et al (<https://www.ncbi.nlm.nih.gov/pubmed/28525758>).

6) Figures S1D-E and Figure S3A show trends but there is no statistical analysis of the data. Given the size of the error bars, especially in S1D-E, it's not clear how robust the differences are, and consequently, how robust the corresponding conclusions are.

7) The authors indicate that Figure 2A suggests that the "plaid pattern" exhibited by this presentation of the data are similar to the "A" and "B" compartments defined by Hi-C experiments. First, more detail describing what is being presented is required. Second, the figure does not define the red or blue regions of the genome that are depicted. Third, the legend indicates this

data is from samples where Pax3 was induced for 1 day. How does this data compare to the day 1 sample in which Pax3 was not induced? There is no mention of samples where Pax3 was induced for 6 days. Are there changes as a function of time? That would seem to be a rather relevant question. If there are changes, what are they and what do they mean?

8) In Figure 2C and related supplementary data, the authors divide Pax3 bound sequences into 4 groups (clusters). These are defined by the histone modifications present around the Pax3 peak. Much later, the reader is told that cluster 1 contains genes essential for myogenesis. It would be helpful if the authors related the four clusters to genes where binding sites correspond to genes and perhaps provide a table, or an extension of a table, indicating which Pax3 bound sequences corresponded to which cluster.

9) In the analysis of the different clusters of Pax3 bound sequences, the authors do not mention that in cluster 4 sites, H3K27Ac and H3K4me3 are induced by Pax3 expression (Fig. 2D). This should be mentioned and explained, if possible.

10a) In Figs. 2E-H and related supplementary information, the authors provide analysis of VISTA enhancers. They report "remarkably, out of 52 enhancers annotated as active in the somites, 10 overlapped with Pax3 ChIP-seq peaks". The reviewer has entirely missed the significance of this analysis. Why is this "remarkable"? Pax3 has long been known to drive expression of genes related to myogenesis – why is it surprising that some enhancers active in somites are bound by Pax3? Data are presented for a subset of the 10 enhancers, but the others are not mentioned. It likely would be interesting to people in the field to see a list of the 10 enhancers bound by Pax3. More importantly, the concluding sentence for this section (page 6) is quite overstated. The authors state that "Pax3 regulates chromatin remodeling of elements displaying an enhancer signature...". The reviewer does not understand how the described pattern of histone modifications at these sequences constitutes "regulation of chromatin remodeling".

10b) The embryo images presented for the VISTA enhancers in Figures 2 and S2 appear to be published at www.enhancer.lbl.gov

If this is true (and I think it is) the fact that these images are not the authors' data should be made explicit in the figure legends.

11) The authors subsequently introduce a newly made line of cells in which inducible Pax3 is tagged. One presumes the line was characterized for its differentiation capabilities and that it differentiates similarly (both qualitatively and quantitatively) to the existing and previously documented cell line. But there is no data presented. A supplemental figure demonstrating the biological integrity of the new line is needed. The line is used for purification and determination of Pax3 interacting proteins, but the text does not reveal when the samples were taken for this experiment. The legend says the silver stain in panel C is from day 4 embryoid bodies. Does that mean day 4 embryoid bodies were used for all aspects of this analysis or just for visualizing the protein? As in point (3) above, does "day 4 embryoid bodies" equate with induction of Pax3 for 4 days?

12) Figure 3 shows that Pax3 is in complex with chromatin looping protein Ldb1 and Smc1. While the IP and STRING analysis are useful, STRING analysis statistics or probability of protein complex presence should be included. What were the calculated confidence scores?

13) The data analysis in Figs 3E-F are incomplete. Co-IP for Pax3 and Ldb1 or Smc1 is presented, but co-localization by IF is limited to Pax3 and Ldb1. Why is the complimentary IF for Pax3 and Smc1 not included? The authors should consider higher resolution images. It would be important to buttress the IP data with the reciprocal IPs or with complimentary data from tissue or perhaps a different myogenic cell line to show that endogenous Pax3 interacts with the looping proteins.

14) The presentation of the data in Fig. 4D needs clarification. The distinction between "Pax3+LDB1 recruit" and "Pax3+Ldb1" is not clearly defined. The Pax3 peak present in "Pax3+Ldb1 recruit" sample is extremely close to the peak in the "Pax3+Ctcf" sample and it is not at all clear how these peaks demarcate domains "A" and "B". Better explanation and perhaps alternate data presentation are required.

15) LDB1 doesn't show binding in cluster 4 (Fig. 4A) but analysis of Pax3+LDB1 peaks (Fig. 4C) shows a change in H3K4me1 deposition to bimodal distribution. Why? Can an explanation be offered?

Reviewer #3:

Remarks to the Author:

Early events of Pax3-driven myogenic differentiation were studied in a mouse ES cell model in order to correlate Pax3 binding with rearrangement of chromatin architecture. Pax3 HiChIP identified subsets of putative enhancers involved in those early differentiation events and these were correlated with appearance of active enhancer histone marks and with activity of some previously characterized VISTA enhancers. Affinity-purification of Pax3-associated proteins identified the co-regulator Ldb1 that was previously associated with looping between LCR and beta-globin genes. Ldb1 ChIP-Seq identified subsets of Pax3-dependent sites that are occupied by Ldb1 but not by Cohesin complex and CTCF. Recruitment of the active enhancer mark H3K4me1 at these subset of enhancers associated with myogenic differentiation appeared to be Ldb1-dependent based on knockdown experiments. These cell culture experiments suggested that Ldb1-dependent mechanism are required for Pax3-driven myogenic differentiation and indeed, cell-specific inactivation of the Ldb1 gene decreased the number of myogenic cells migrating into mouse forelimb buds during development. Hence, Ldb1 appears critical for completion of the myogenic program.

The use of the Pax3 HiChIP to reveal the subset of newly formed contacts during differentiation is powerful and an efficient way to identify critical subsets of enhancers. Further, the association of this subset with Ldb1 recruitment, independently of Cohesin/CTCF presence, suggests a unique mechanism for subTAD interactions. Both aspects are thus novel and will be of interest not only for those in the myogenic field but more broadly. It will be interesting to see whether this unique role of Ldb1 is limited to a few transcription factors or whether it is a more general mechanism. Irrespective, the association of Ldb1 with Pax3-driven changes in chromosome architecture during differentiation is novel and important.

The authors have supported the importance of Ldb1 for myogenic differentiation (Fig. 5 C-F) and for recruitment of active enhancer marks at enhancers (Fig. 5B) but not for maintenance of long-range interactions. It would be interesting, if possible by a PCR approach, to assess a representative subset of those interactions in the shLdb1 system used in Fig. 5B in order to correlate enhancer activation with physical association.

Reviewer #4:

Remarks to the Author:

In this manuscript, Magli et al. describe the role of Ldb1 and Pax3 in chromatin interactions during lineage specification. The figures are well prepared and the manuscript is well written. However, there are a few major issues with the manuscript.

1. My most major issue is that Magli et al says that Ldb1 and Pax3 are involved in chromatin interactions. However, there are no perturbation experiments that they do to show this. All the

work is based on correlations - on chromatin interactions found at Pax3 binding sites. The authors need to perform siRNA or shRNA or small molecule inhibitor experiments and then examine the chromatin interactions in order to say that the chromatin interactions are indeed mediated by particular molecules. I understand that it will be difficult to do HiChIP on PAX3 after Pax3 has been knocked down, however the authors can perform 3C-qPCR or 4C or Hi-C and see whether the chromatin interactions are perturbed.

2. Similarly, there is no evidence that Pax3-mediated chromatin interactions are involved in lineage specification, only a correlation. The authors need to perturb chromatin interactions (e.g. Deng et al., Cell, 2012 or Fanucchi et al., Cell 2013) in order to show that without such interactions, lineage specification is indeed disrupted.

3. How do Pax3 and Ldb1 work together with the cohesin components in the complex?

Reviewer #1

This study investigates the role of transcription factors in remodelling chromatin during commitment of myogenic cells. Using HiChIP and mass spectrometry assays of Pax3 interacting partners, the authors identify the widely expressed Ldb1 transcription factor as a critical component in Pax3-mediated chromatin looping during myogenic commitment in an in vitro differentiation model. They subsequently perform analysis and specific gene KO experiments in vivo to demonstrate that Ldb1 is critical for maintaining the number of Pax3+ cells in the developing mouse limb bud. The results and conclusions of the study provide a significant advancement of our understanding of how chromatin is regulated in multiple steps via an interplay between general regulators (Ctcf, cohesion) and transcription factors during myogenic cell commitment. There are a few concerns that the authors need to address to clarify and improve the study. These are listed below.

One point concerns the title – the word "reprograms" might be out of context here. Perhaps "modifies" or "directs", or some similar word would be more appropriate. Also, some functional data is provided for Ldb1, so indicating this TF in the title is suggested.

We appreciate the positive and constructive feedback provided by this Reviewer. As suggested, we have revised the title to: "Pax3 cooperates with Ldb1 to direct local chromosome architecture during lineage specification"

Specific Comments:

1) Some more information on the protocol for induction and differentiation is needed to understand the homogeneity of the cells being analysed and the differentiation scheme. In the methods, Pax3 was induced in 3day EBs and analysis was done at Days 1 and 6. The scheme indicates ES and mesoderm precursors. It would be more accurate to indicate EBs instead of "mesoderm precursor". Also, the authors induce differentiation using a Dox

system of Pax3 expression. Please show staining of the cells at Day 1 and Day 6 (assay points; also pre-assay points as controls) for Pax3, Myf5, MyoD and Myogenin staining. The detection of Myf5 transcripts at Day 1 is not a sufficient indicator of what is happening at the single cell level and the dynamics of the state of the cells using this protocol. The authors need to indicate these percentage values for the assay times used.

We thank the Reviewer for bringing this up. In revised Supplementary Figure 1, we now show the characterization of the myogenic differentiation induced by Pax3. We show immunostaining demonstrating PAX3 expression, western blot for MYOD and MYOG in non-induced, 1-day and 6-day induced cells, and flow cytometry analysis for MYOD and MYOG at both time points. We were not able to include staining for MYF5 as the reliable antibody we used in revised Fig. 7a, is no longer commercially available, and unfortunately, we were not able to find a suitable reliable alternative. Nevertheless, levels of Myf5 transcript at these time points are reported in revised Supplementary Figure 2b.

2) Please explain why it is expected that Pax3 occupies -111 and -57 positions in the Myf5 locus. It is not clear what these elements are doing, what they represent, and if Pax3 binds these sites in the embryo.

Myf5 is a classic and well-studied Pax3 target gene and the -111kb and -57kb enhancers are bound by this transcription factor during embryogenesis, thus serving as positive controls for our ChIP experiments. We have clarified this aspect in the manuscript and included the appropriate references.

3) The Rudnicki lab reported genome-wide analysis of Pax3 and Pax7 in adult myogenic cells. It would be informative to compare that data with the one presented in this study.

We appreciate this suggestion, but this is the focus of another study currently in preparation, which focuses on myogenic progenitor cells (as opposed to mesoderm patterning, the topic of this study). In that paper, we dissect the transcriptional changes induced by Pax3 in embryonic and adult skeletal muscle progenitor cells, as well as in fibroblasts. As this is a point of information rather than central to the mechanistic studies in the current paper, and is indeed addressed in our upcoming paper on myogenic progenitor cells, we believe the comparison of these datasets is not central for the main conclusion of the present manuscript. We hope the reviewer agrees with our point of view.

4) Page 4: " peaks are associated with genes located many kilobases away ". Do the authors mean sequences or genes? Away from what landmark?

We thank the Reviewer for allowing us to clarify this sentence. We have revised the text, as follows: ...we observed that the vast majority (~90%) of Pax3 ChIP-seq peaks are more than 5 kilobases (>5kb) away from the transcription start site (TSS) of the nearest gene.

5) Page 4: the authors give as examples " Gas1 +29kb, Megf10 +135kb, Meis2 +144kb, Myf5 -111kb, MyoD -27kb and Vcam1 +3kb" for long range interactions. It is not clear what exactly constitutes long-range, but "+3kb" might not fit into this definition. The authors need to clarify for the general reader how these distinctions are made.

As mentioned in the response above (#4), approximately 90% of Pax3 binding sites are located more than 5kb away from the TSS of the closest annotated gene. To avoid confusion for the general

reader, we clarified this point in the text and provided the average size of long range interactions detected using HiChIP in revised Fig 1d.

6) Figure 1B: there appears to be a strong peak and a couple of smaller peaks for Pax3 in the Myf5 gene at Day 1 but not at Day 6. Please comment on the significance of this, as it might be expected to be the other way around.

Both ChIP-seq (revised Fig. 1b) and ChIP-qPCR (revised Supplementary Figure 1e) show binding of Pax3 to *Myf5* upstream regions. We detected also peaks in the *Myf5* gene, as the reviewer noted, but they are not consistent among replicates. Other investigators have referred to such peaks as “indirect peaks” (<https://www.ncbi.nlm.nih.gov/pmc/articles/PMC4198380/>), which resulted from chromatin looping events, which are not directly bound by the TF. We believe the Pax3 peak may represent such an event.

7) Figure 2A: please indicate why Chr12 was chosen? Are Pax3 targets more enriched on this chromosome? Chr10 contains Myf5, would this have been a better choice for analysis? Also in Figure 2B, E, G, please explain the significance of the different Histone marks marking the Pax3+ enhancers.

This panel has been removed from the revised manuscript. This was replaced with a snapshot of the chromatin marks at the *Myf5* locus.

8) Please explain the significance of the Histone modifications for the general reader and their relationships amongst each other, as well as annotations of Figures (ex. Red box = 9 Figure 2B). For example, Fig S2A, there are correlative and anti-correlative (Cluster 2 and 3) peaks for K4me1 and K27ac in the examples given – why is this the case if both should mark enhancers.

We have clarified this aspect in the text and included an additional reference for the general reader (<https://www.ncbi.nlm.nih.gov/pubmed/29122461>). Cluster 2 includes elements proximal to TSS (high H3K4me3 and H3K27Ac). Cluster 3 represents repressed elements (high H3K27me3).

9) Please indicate in all Figures the source of the cells and use uniform timing scheme throughout. Ex. Fig 4B indicates 4 day EBs – which correlates to Day 1. What is the timing for the other experiments, and their n-values. A supplementary Figure on replicates is provided for some experiments, but this is not obvious for all of the experiments in the study. All the n-values need to be indicated in the legends. Similarly, how was the data calculated in Figure 5 for Pax3 and MYHC stainings? Although 3 independent embryos were analysed, how many sections were used, and how was the area calculated? This is a rather imprecise quantification method for Pax3 given the nuclear staining, however it might be ok for MYHC quantification.

We thank the Reviewer for pointing out the inconsistency in the timing scheme and we apologize for any confusion. We have corrected this aspect throughout the text. Results reported in this manuscript are representative of independent biological triplicates, now indicated in each figure legend. Sequencing data were performed on independent biological duplicates with the exception of the H3K4me1 ChIP seq shown in Figure 6, which was validated on 3 biological replicates by ChIP-qPCR. Analysis of embryo sections was carried using 2 images for each embryo. The analysis was performed using ImageJ (also called Fiji) and focused on measuring the area positive to a given fluorescent signal. This type of analysis is commonly used for example in other

published studies such as <https://www.ncbi.nlm.nih.gov/pubmed/19066593>. By normalizing to the number of nuclei present in each image, we provide a measurement independent on the field of view. For the purpose of this analysis, we consider equivalent both the nuclear and the cytoplasmic signals.

10) The example provided in Figure 5C is too faint to assess what is happening to Myf5 staining. Also, Fig S4B, the "nuclei" panel does not contain only nuclei.

As suggested by this Reviewer, we included an image with a more pronounced Myf5 staining in the forelimb area of these embryos in revised Figure 7a. We adjusted the brightness and contrast of the signal associated with Myf5 staining in order to better adapt it to the intensity distribution of the data.

11) Indicate more clearly in Figure legends for 4D and F what is the difference between category 1 and 2.

For a simpler interpretation of the data, we are now focusing only on the group of loci characterized by Pax3-mediated Ldb1 recruitment, which we refer as “Pax3+Ldb1” sites.

12) Although the authors focused their study on the embryo, Pax3 is largely downregulated in the mouse fetus. It would be interesting to determine if the same principles proposed for gene regulation apply to Pax7. It is not necessary to perform a detailed study as has been done, but nonetheless, it would be highly informative and complementary to provide a wider message to readers, as a paradigm-principle for the model that the authors propose, and that is not restricted only to analysis of embryonic cells.

We thank the reviewer for this interesting suggestion. We agree with this Reviewer about the potential broad significance of Ldb1 function in myogenic progenitors. Accordingly, we have performed co-immunoprecipitation using anti-LDB1 antibody in 1-day induced Pax7 embryoid bodies, and our data showed that, similar to Pax3, Pax7 can also interact with Ldb1 (revised Supplementary Figure 5d).

Reviewer #2

The manuscript submitted by Magli et al is interesting and contains novel information about Pax3 and Ldb1 but is premature for publication in its current form. Perhaps the most notable problems are that a main conclusion is not adequately supported and the genome-wide data sets are not adequately interrogated or presented. In addition, it appears that the different assays were performed at different times, raising questions about the validity of the data comparisons.

We thank this Reviewer for recognizing the novelty of our findings. In the revised manuscript, we expanded our analyses and provided additional experimental data to support our conclusions. We hope the Reviewer finds the revised version of the manuscript satisfactorily addresses these concerns.

1) A major conclusion of the work is that the Ldb1 protein is “the mediator of looping interactions at Pax3 binding sites” (quoted from abstract). The Ldb1 depletion studies nicely show biological relevance in myogenesis, but the connection to looping is limited to decreased deposition of H3K4me1 at sites occupied by Ldb1 and Pax3. Such a result simply does not

show Ldb1 mediates looping, and is it important to note that even though the data (Fig. 5B) are statistically significant, the decrease in deposition of H3K4me1 at the sites is less than 2-fold in every instance. It is hard to understand how a very modest decrease in this histone modification leads to the aforementioned conclusion. The clearest experiment to address the issue is to compare the Hi-ChIP results in control and Ldb1 knockdown cells.

We thank the Reviewer for this suggestion. We now report the data for Pax3 HiChIP from control and Ldb1 knockdown cells (revised Figure 5). On two biological replicates, we consistently found decreased numbers of contacts at Pax3+Ldb1 sites as well as decreased numbers of loops. In contrast, the number of loops involving Pax3 only sites is not affected by Ldb1 knockdown. This strongly supports our model. We agree with the Reviewer that these data clearly address the raised issue, and accordingly, have been included in the manuscript.

2) The authors also correlate loop formation between Pax3 bound sequences with gene expression, but the study lacks direct evidence that loop formation is required for transcription.

To address this aspect, we now report data demonstrating that knockdown of Ldb1 in iPax3 cells impairs myogenic gene expression (revised Fig. 5c). In addition, we generated a series of fusion proteins to comprehensively study the function/necessity of candidate Pax3 cofactors identified by mass spectrometry. To target these candidate cofactors to Pax3-bound elements, we used the DNA binding domain of Pax3 (referred as Δ TAD) which: 1) retains DNA binding activity (revised Supplementary Fig 7b); 2) is devoid of myogenic activity (revised Fig. 6d); 3) is incapable of recruiting Ldb1 at Pax3 sites (revised Fig 6c and revised Supplementary Fig. 7a); and 4) does not induce increase in H3K4me1 at Pax3 bound elements (revised Fig 6f-h and revised Supplementary Fig. 7e-g). These features make the Δ TAD a perfect tool for studying the effect of protein targeting to Pax3 bound sites. As shown in revised Fig. 6, our results demonstrate that the Δ TAD-LDB1 fusion protein recapitulates PAX3 WT myogenic commitment functions. This shows that recruiting LDB1 is sufficient for this activity of Pax3. We believe this makes a strong argument in favor of the importance of Ldb1 in the Pax3-mediated transcriptional regulation.

3) The authors present multiple genome-wide studies but fail to present the results, preferring to pick out specific loci or other specific information and solely present that.

a) Pax3 ChIP-seq: No information is given on the number of reads and number of mapped reads. The number of identified peaks is not given. Was it 20 or 20,000? Could a supplemental table be presented identifying the binding sites? Are there distinguishing differences in Pax3 binding between day 1 and day 6 samples, and if so, what are they? What is the relationship between the location of Pax3 binding sites and genes? Importantly, the authors conclude that most Pax3 binding sites are distant from genes. What determines whether a binding site is distant from a nearby gene vs a binding site being intergenic? Perhaps comparison of Pax3 binding sites to regions of RNA pol II binding might better inform the relationship between Pax3 localization and gene expression.

We thank the Reviewer for bringing this up. In revised Fig. 1c, we now report number and annotation for Pax3 ChIP-seq data in mesodermal cells (1-day) and myogenic progenitors (6-day). As mentioned in the response above for Reviewer 1 (point #3), we have another manuscript in preparation, which focuses on a detailed analysis of the loci bound by Pax3 in gene-centric point of view. We believe these additional studies are beyond the central focus of the present manuscript, which is the interaction of Pax3 with Ldb1 and chromatin looping. We hope this Reviewer understands our point of view.

b) The same concerns and questions apply to the Ldb1 and Smc1 ChIP-seq data. It is simply not appropriate to do a genome wide study and then provide no data or analysis whatsoever except for the very small subset of the data that relates to Pax3 binding.

The Reviewer is correct. We provide additional data in this regard in revised Supplementary Fig. 6a.

c) Hi-ChIP: No information is given about the number of interactions observed across the genome and whether the number of observed interactions is evenly distributed amongst Pax3 binding sites or whether some Pax3 binding sites are “hotspots” for interactions. Is there bias in the frequency of interactions based on chromosome (ie – are interactions more frequent on specific chromosomes or on “gene-rich” chromosomes)? Is there a way global interactions could be summarized in tabular or graphical form? Would it be too much to confirm a few of the selected interactions by conventional 3C?

Thanks for the suggestion. We now provide the requested data regarding our HiChIP experiments in a tabular form in revised Fig. 1d and revised Fig. 5f.

4) There are disconnects in the data comparisons. It appears the Pax3 ChIP-seq and the Hi-ChIP are comparisons of day 1 and day 6 post induction of Pax3. it seems all of the other ChIP-seq analyses of histone modifications are from day 4 embryo bodies. Does this correspond to 4 days post-induction of Pax3? The legend to Figure 2C suggests that this is the case. So why is it valid to compare histone ChIP-seq from day 4 to Pax3 ChIP-seq and Hi-ChIP on days 1 and 6? In Figure S3B, the authors present RNA-seq from day 5. How does this relate to the other studies done on different days?

We apologize for any confusion due to the use of different nomenclature for the time points. We rectified this throughout the manuscript by referring to non-induced, 1-day and 6-day induced for both ChIP-seq and HiChIP. In addition, we also used previously published RNAseq data (Magli A et al, 2014) generated following a 2-day Pax3 induction to demonstrate that Pax3+Ldb1 loci can be annotated to differentially expressed genes (revised Supplementary figure 6c). These data are in agreement with our 1-day qPCR data shown in revised Supplementary Fig. 2b and are supportive of our conclusions.

In addition, the authors consistently use the phrase “chromatin remodeling” when talking about the chromatin landscape defined by the histone modifications that were assayed. Many (most?) people use “chromatin remodeling” to talk about ATP-dependent enzymatic alterations to chromatin and use chromatin or histone “modifications” to talk about the various chemical groups that can be added and removed from the histones. The authors should consider this point and justify their use of “chromatin remodeling” or change the text throughout.

We understand the Reviewer’s point, and accordingly we revised the text throughout by referring to histone post-translational modification or epigenetic marks.

5) In Figure 1D, the authors state that long-range interactions are clearly defined for specified genomic loci. However, it is not possible to associate the regions indicated with the density matrices. The authors should consider alternate presentations of the interactions, possibly in the manner of data presented in other Hi-ChIP papers such as Mumbach et al

(<https://www.ncbi.nlm.nih.gov/pubmed/27643841>) or Nora et al
(<https://www.ncbi.nlm.nih.gov/pubmed/28525758>).

We revised data presentation as suggested by this Reviewer. In addition, we now provide matrices generated following comparison with the Pax3 HiChIP from non-induced cells, which represents our negative control.

6) Figures S1D-E and Figure S3A show trends but there is no statistical analysis of the data. Given the size of the error bars, especially in S1D-E, it's not clear how robust the differences are, and consequently, how robust the corresponding conclusions are.

All experiments are independent biological replicates and statistical analyses are provided for all the significant differences.

7) The authors indicate that Figure 2A suggests that the “plaid pattern” exhibited by this presentation of the data are similar to the “A” and “B” compartments defined by Hi-C experiments. First, more detail describing what is being presented is required. Second, the figure does not define the red or blue regions of the genome that are depicted. Third, the legend indicates this data is from samples where Pax3 was induced for 1 day. How does this data compare to the day 1 sample in which Pax3 was not induced? There is no mention of samples where Pax3 was induced for 6 days. Are there changes as a function of time? That would seem to be a rather relevant question. If there are changes, what are they and what do they mean?

For clarity, in the revised manuscript we have replaced this panel with the snapshot of the chromatin marks at the *Myf5* locus. In addition, as mentioned in response #5 above, we also performed Pax3 HiChIP in non-induced cells and compared this sample to 1-day and 6-day Pax3 HiChIP data to better illustrate the Pax3-induced changes.

8) In Figure 2C and related supplementary data, the authors divide Pax3 bound sequences into 4 groups (clusters). These are defined by the histone modifications present around the Pax3 peak. Much later, the reader is told that cluster 1 contains genes essential for myogenesis. It would be helpful if the authors related the four clusters to genes where binding sites correspond to genes and perhaps provide a table, or an extension of a table, indicating which Pax3 bound sequences corresponded to which cluster.

Thank you for this suggestion. We used k-means clustering for multiple comparisons. We included specific reference to each cluster in the relative Figure to avoid further confusion.

9) In the analysis of the different clusters of Pax3 bound sequences, the authors do not mention that in cluster 4 sites, H3K27Ac and H3K4me3 are induced by Pax3 expression (Fig. 2D). This should be mentioned and explained, if possible.

We thank the Reviewer for bringing this up. As suggested, we mention this observation in the revised manuscript. Increased deposition of H3K4me3 is the result of the interaction with the HMT complex, which we detected in the Pax3 interactome. Other investigators, as in the case of Lee et al (<https://www.ncbi.nlm.nih.gov/pubmed/24368734>), have performed an *in vitro* methylation assay using purified SET1A/B and MLL3/4 complexes, and in both cases, the specificity toward K4me3 and K4me1 was not absolute. We believe this may explain the increase in H3K4me3 observed at sites characterized by clear increase in H3K4me1. Regarding H3K27Ac, our data show

no interaction with p300 and CBP histone acetyltransferase. We believe recruitment of these enzymes follows a different dynamic.

10a) In Figs. 2E-H and related supplementary information, the authors provide analysis of VISTA enhancers. They report “remarkably, out of 52 enhancers annotated as active in the somites, 10 overlapped with Pax3 ChIP-seq peaks”. The reviewer has entirely missed the significance of this analysis. Why is this “remarkable”? Pax3 has long been known to drive expression of genes related to myogenesis – why is it surprising that some enhancers active in somites are bound by Pax3? Data are presented for a subset of the 10 enhancers, but the others are not mentioned. It likely would be interesting to people in the field to see a list of the 10 enhancers bound by Pax3. More importantly, the concluding sentence for this section (page 6) is quite overstated. The authors state that “Pax3 regulates chromatin remodeling of elements displaying an enhancer signature...”. The reviewer does not understand how the described pattern of histone modifications at these sequences constitutes “regulation of chromatin remodeling”.

We included the VISTA data to support the notion that our Pax3 bound elements correspond to enhancers active during embryogenesis. Although this can be obvious from a developmental biology point of view, these results validate the use of differentiating mouse ES cells to study the molecular events driving myogenic commitment. To avoid overstatements, we revised the text referred above by this Reviewer. After reanalysis of the ChIP-seq, using more stringent parameters, we identified 9 elements overlapping to VISTA enhancers active in the somites, and we provide the list in the revised Supplementary Fig. 4b.

10b) The embryo images presented for the VISTA enhancers in Figures 2 and S2 appear to be published at www.enhancer.lbl.gov. If this is true (and I think it is) the fact that these images are not the authors’ data should be made explicit in the figure legends.

The use of images from the VISTA enhancer catalog (with associated reference) was clearly stated in the main text as well as material & methods of the original manuscript. To make this more explicit, the website reference is now also included in the respective figure legends.

11) The authors subsequently introduce a newly made line of cells in which inducible Pax3 is tagged. One presumes the line was characterized for its differentiation capabilities and that it differentiates similarly (both qualitatively and quantitatively) to the existing and previously documented cell line. But there is no data presented. A supplemental figure demonstrating the biological integrity of the new line is needed. The line is used for purification and determination of Pax3 interacting proteins, but the text does not reveal when the samples were taken for this experiment. The legend says the silver stain in panel C is from day 4 embryo bodies. Does that mean day 4 embryo bodies were used for all aspects of this analysis or just for visualizing the protein? As in point (3) above, does “day 4 embryo bodies” equate with induction of Pax3 for 4 days?

Cell line characterization is provided in the revised Supplementary Fig. 5a-b. As mentioned above, we apologize for the confusion created with the nomenclature of time points used for analysis. We revised the text to “1-day” induction throughout the manuscript.

12) Figure 3 shows that Pax3 is in complex with chromatin looping protein Ldb1 and Smc1. While the IP and STRING analysis are useful, STRING analysis statistics or probability of protein complex presence should be included. What were the calculated confidence scores?

We provide all the mass-spectrometry data and STRING scores in Supplementary Table 1.

13) The data analysis in Figs 3E-F are incomplete. Co-IP for Pax3 and Ldb1 or Smc1 is presented, but co-localization by IF is limited to Pax3 and Ldb1. Why is the complimentary IF for Pax3 and Smc1 not included? The authors should consider higher resolution images. It would be important to buttress the IP data with the reciprocal IPs or with complimentary data from tissue or perhaps a different myogenic cell line to show that endogenous Pax3 interacts with the looping proteins.

In revised Supplementary Figure 5, we now show immunofluorescence staining for PAX3 and SMC1 in E9.5 embryo as well as the reciprocal co-IP using anti-LDB1 antibody. Image resolution may be affected by pdf compression from the .eps format. We will ensure high resolution images will be provided to the editorial team.

14) The presentation of the data in Fig. 4D needs clarification. The distinction between “Pax3+LDB1 recruit” and “Pax3+Ldb1” is not clearly defined. The Pax3 peak present in “Pax3+Ldb1 recruit” sample is extremely close to the peak in the “Pax3+Ctcf” sample and it is not at all clear how these peaks demarcate domains “A” and “B”. Better explanation and perhaps alternate data presentation are required.

For a more straightforward interpretation of the data, we are now focusing only on the group of loci characterized by Pax3-mediated Ldb1 recruitment, which we refer as “Pax3+Ldb1” sites. We removed the section relative to Pax3+Ctcf peaks from the text.

15) LDB1 doesn't show binding in cluster 4 (Fig. 4A) but analysis of Pax3+LDB1 peaks (Fig. 4C) shows a change in H3K4me1 deposition to bimodal distribution. Why? Can an explanation be offered?

Our H3K4me1 ChIP seq data comparing Pax3 WT and Δ TAD-LDB1 (revised Fig 7) shows that only a subset of elements is characterized by an increase in H3K4me1 following Δ TAD-LDB1 induction, which also induces skeletal myogenesis. This suggests that different mechanisms participate in the recruitment of the HMT complex at Pax3 sites. Interestingly, increased H3K4me1 resulting from forced targeting of the HMT complex to these elements, is not sufficient to activate the myogenic program. Future experiments will clarify the interplay between these protein complexes at Pax3 binding sites.

Reviewer #3

Early events of Pax3-driven myogenic differentiation were studied in a mouse ES cell model in order to correlate Pax3 binding with rearrangement of chromatin architecture. Pax3 HiChIP identified subsets of putative enhancers involved in those early differentiation events and these were correlated with appearance of active enhancer histone marks and with activity of some previously characterized VISTA enhancers. Affinity-purification of Pax3-associated proteins identified the co-regulator Ldb1 that was previously associated with looping between LCR and beta-globin genes. Ldb1 ChIP-Seq identified subsets of Pax3-dependent sites that are occupied by Ldb1 but not by Cohesin complex and CTCF. Recruitment of the active enhancer mark H3K4me1 at these subset of enhancers associated with myogenic differentiation appeared to be Ldb1-dependent based on knockdown experiments. These cell culture experiments suggested that Ldb1-dependent mechanism are required for Pax3-driven myogenic differentiation and indeed, cell-specific inactivation of

the Ldb1 gene decreased the number of myogenic cells migrating into mouse forelimb buds during development. Hence, Ldb1 appears critical for completion of the myogenic program. The use of the Pax3 HiChIP to reveal the subset of newly formed contacts during differentiation is powerful and an efficient way to identify critical subsets of enhancers. Further, the association of this subset with Ldb1 recruitment, independently of Cohesin/CTCF presence, suggests a unique mechanism for subTAD interactions. Both aspects are thus novel and will be of interest not only for those in the myogenic field but more broadly. It will be interesting to see whether this unique role of Ldb1 is limited to a few transcription factors or whether it is a more general mechanism. Irrespective, the association of Ldb1 with Pax3-driven changes in chromosome architecture during differentiation is novel and important.

The authors have supported the importance of Ldb1 for myogenic differentiation (Fig. 5 C-F) and for recruitment of active enhancer marks at enhancers (Fig. 5B) but not for maintenance of long-range interactions. It would be interesting, if possible by a PCR approach, to assess a representative subset of those interactions in the shLdb1 system used in Fig. 5B in order to correlate enhancer activation with physical association.

We really appreciate the enthusiasm of this reviewer for our work, and we are happy to provide further data regarding Ldb1 function. As described above (Reviewer 2, #1), we performed Pax3 HiChIP in control and knockdown cells, and this resulted in decreased numbers of detected loops at sites characterized by Pax3-dependent Ldb1 recruitment (please see revised Fig. 5d-f). The requirement of Ldb1 in this process is further confirmed by our studies using Δ TAD-LDB1, a fusion protein between the Pax3 DNA binding domain (which provides only targeting to Pax3 sites) and full-length LDB1. This protein, but not Δ TAD-ASH2L, is able to recapitulate Pax3 myogenic functions and, more interestingly, is able to induce an increase in H3K4me1, thus suggesting potential recruitment of the HMT complex.

Reviewer #4

In this manuscript, Magli et al. describe the role of Ldb1 and Pax3 in chromatin interactions during lineage specification. The figures are well prepared and the manuscript is well written. However, there are a few major issues with the manuscript.

We thank this Reviewer for the nice comments.

1. My most major issue is that Magli et al says that Ldb1 and Pax3 are involved in chromatin interactions. However, there are no perturbation experiments that they do to show this. All the work is based on correlations - on chromatin interactions found at Pax3 binding sites. The authors need to perform siRNA or shRNA or small molecule inhibitor experiments and then examine the chromatin interactions in order to say that the chromatin interactions are indeed mediated by particular molecules. I understand that it will be difficult to do HiChIP on PAX3 after Pax3 has been knocked down, however the authors can perform 3C-qPCR or 4C or Hi-C and see whether the chromatin interactions are perturbed.

We thank the Reviewer for the comment and suggestion. Accordingly, we have performed HiChIP using Pax3 antibody in control and Ldb1-knockdown cells. As shown in revised Fig 5d-f, we detected a decreased numbers of loops following Ldb1 knockdown. Moreover, we included an additional control in the HiChIP experiments reported in revised Fig. 1 by performing this procedure in the absence of Pax3-induction. We used this dataset as the negative control for our HiChIP, and we report the interaction matrices as fold +dox vs. no dox. This perturbation

experiment confirms that Pax3 function is impaired in absence of Ldb1, as evidenced by decreased myogenic differentiation and target gene expression (Revised Figure 5a).

2. Similarly, there is no evidence that Pax3-mediated chromatin interactions are involved in lineage specification, only a correlation. The authors need to perturb chromatin interactions (e.g. Deng et al., Cell, 2012 or Fanucchi et al., Cell 2013) in order to show that without such interactions, lineage specification is indeed disrupted.

To address this point, we took advantage of a series of fusion proteins involving the Pax3 DNA binding domain (referred as Δ TAD) and LDB1, ASH2L, CXXC1 and PAGR1A, respectively. As shown in revised Fig. 6 and revised Supplementary Fig. 7, only Δ TAD-LDB1 is capable of recapitulating Pax3 WT function. We believe these experiments highlight the instrumental function of Ldb1 in Pax3-mediated myogenic specification.

3. How do Pax3 and Ldb1 work together with the cohesin components in the complex?

Based on our findings, Pax3 does not recruit SMC1 at its binding sites. While Pax3-dependent Ldb1 is evident at cluster 1 loci (revised Fig 4a), it is possible the PAX3-SMC1 interaction what we see by Co-IP (revised Fig 3e) may be dependent on LDB1 interaction with CTCF. Accordingly, LDB1-CTCF interaction has been described by the group of Ann Dean at NIH (Lee J et al, 2017) and approximately 20% of LDB1 peaks detected in our ChIP-seq experiments overlap with CTCF or SMC1 peaks (revised Supplementary figure 6a). Future studies will clarify the interplay between Ldb1 and CTCF-Cohesin.

Thank you for your consideration, and we hope that our extensive new set of data address all of the concerns. I look forward to your response.

Reviewers' Comments:

Reviewer #1:

Remarks to the Author:

Magli et al.

Pax3 reprograms local chromosome architecture during lineage specification

The authors have for the most part satisfactorily answered the queries raised and have significantly improved the manuscript by providing additional experiments and revising the manuscript. There are some outstanding points that need clarification.

1) For many of the gene annotations (Megf10, Met, Myf5, etc.; ex. Figs 1 and 2) it is unclear where is the gene body, and where is the promoter of the gene. This information is necessary to better appreciate the location of the ChIP peaks.

2) Response to previous point #3 regarding data of Pax3 ChIP from the Rudnicki lab. The reviewer appreciates that the authors would like to address this in another manuscript. Nevertheless, some minimal comparisons would be important to make here to assess which Pax3 peaks corresponding to the genes analysed here correlate with the previous analysis. Otherwise, the reader is left with a knowledge gap that begs response in the present study. Perhaps the authors could report some aspects related directly to the analysis done here without taking too much from the other manuscript in preparation.

Reviewer #2:

Remarks to the Author:

This reviewer was Reviewer 2 in the original review.

The revised manuscript by Magli et al addresses the minor comments in the first review but despite the data added, it does not adequately address two major issues raised in the original review. In addition, the third critique of the original review pointed out that the authors simply extracted a limited amount of information relevant to the points of this manuscript from genome-wide ChIP-seq data sets without providing a general description or any general analyses of those ChIP-seq data sets. The authors response was to provide a limited amount of data to address some of the specific questions listed in the third critique of the original review and to decline to provide any additional general information about the datasets because a different manuscript that will provide this information is in preparation.

Original Critique #1:

At issue are the examples of changes in short- and long-range interactions due to Ldb1 knockdown shown in Figure 5g. The authors concluded that "visual inspection of the HiChIP matrices confirmed the reduced long-range interactions at loci characterized by Pax3-dependent Ldb1 recruitment (Dbx1 and Cdon loci in Fig. 5g)", but the data is not convincing and seems to indicate a qualitative change in the location of interactions, not a loss of interactions. A clear decrease in the number of interactions is not apparent despite the table indicating that ~190 loops are lost with Ldb1 knockdown. Is there a better way to convey these findings?

Original critique #2:

The point of this critique was that the authors had not directly connected looping between Pax3 bound sequences and expression of those genes. The response to the critique was description of an experiment where the DNA binding domain of Pax3, which does not have any additional functions, was fused to Ldb1. The fusion protein recapitulates the myogenic properties of wildtype Pax3. While this experiment, as the authors state in the rebuttal letter, "shows that recruiting LDB1 is sufficient for the activity of Pax3", it does not demonstrate that looping mediates

transcription. It demonstrates that targeting Ldb1 to Pax3 bound loci mediates the function of Pax3. The authors could have attempted to determine the requirement for specific Pax3 sites in loop formation and in mediating gene expression at any myogenic locus by manipulating endogenous sequences or even by re-creating a loop using myogenic sequences in a reporter assay of some sort. Alternatively, the Hi-ChIP data from the control and Ldb1 knockdown cells could be mined to identify loops at myogenic genes. If the loops were altered and/or less frequent in the knockdown cells, this would at least extend the correlation between looping and myogenic gene expression, and the authors could modify their conclusions to indicate that a correlation exists.

Original critique #3 (also relevant to critique #8):

In response to the request for results and analysis of the Pax3 ChIP-seq data set, the authors provided the number of peaks on days 1 and 6 and the % of peaks in different genomic locations. The data provide assurance that the conclusions are not based on small number of peaks, but general features of the data set, both technical and biological, remain unknown. For example, a major conclusion of the manuscript is that Pax3 targeting of Ldb1 promotes myogenesis, but the authors do not provide information about the locations of Pax3 binding sites relative to myogenic genes across the genome. So how can the reviewer evaluate the conclusion?

Similar critiques were made about the Ldb1 and Smc1 ChIP-seq datasets. The authors' response was to provide Fig. S6a, which features a Venn diagram comparing the number of peaks and numbers of overlapping peaks at day 1 in induced and uninduced cells. This provides no specific information about the differences between the induced and uninduced state or any information about the locations/identities of these peaks. It is a collection of numbers that is of limited biological meaning.

A manuscript "in preparation" could be a year or two from publication. This reviewer finds the concept of generating and using a genome-wide data set but reserving key features and results for an undetermined amount of time to be unacceptable. If there is not a journal policy that all data needs to be presented, there should be.

Minor point:

Line 102 of the text refers to Fig. S1D. Did the authors mean to refer to Fig. S2B?

Reviewer #3:

Remarks to the Author:

The authors have adequately dealt with this reviewer's queries.

Reviewer #4:

Remarks to the Author:

I am satisfied with the revision and have no further questions.

Reviewer #1:

The authors have for the most part satisfactorily answered the queries raised and have significantly improved the manuscript by providing additional experiments and revising the manuscript. There are some outstanding points that need clarification.

We thank this reviewer once again for the positive feedback.

1) For many of the gene annotations (Megf10, Met, Myf5, etc.; ex. Figs 1 and 2) it is unclear where is the gene body, and where is the promoter of the gene. This information is necessary to better appreciate the location of the ChiP peaks.

We understand this reviewer's concern. Nevertheless, because most of the regions presented in this manuscript span several kilobases, some of the annotation elements would be too small if included. However, to address this point, we have included arrows to indicate the location of the transcription start site and used a different color to specify the intergenic regions.

2) Response to previous point #3 regarding data of Pax3 ChiP from the Rudnicki lab. The reviewer appreciates that the authors would like to address this in another manuscript. Nevertheless, some minimal comparisons would be important to make here to assess which Pax3 peaks corresponding to the genes analysed here correlate with the previous analysis. Otherwise, the reader is left with a knowledge gap that begs response in the present study. Perhaps the authors could report some aspects related directly to the analysis done here without taking too much from the other manuscript in preparation.

As mentioned above, the manuscript describing the comparison of Pax3 binding in differentiating ES cells (1-day and 6-day induction) and myoblasts (Rudnicki lab) has been recently accepted for publication in PLOS Biology. Based on our analyses, there is a partial overlap in the subset of loci bound by Pax3 among these cell types, which is in agreement with the cell type-dependent genomic occupancy reported for other transcription factors (<https://www.ncbi.nlm.nih.gov/pubmed/29358654>). As requested by this reviewer, we have revised the results section to clarify the cell-type changes in Pax3 genomic binding. Importantly, these data highlight the usefulness of the mouse ES cell-based differentiation system for studying the molecular mechanisms behind Pax3-dependent skeletal myogenic commitment from nascent mesoderm.

Reviewer #2:

This reviewer was Reviewer 2 in the original review.

The revised manuscript by Magli et al addresses the minor comments in the first review but despite the data added, it does not adequately address two major issues raised in the original review. In addition, the third critique of the original review pointed out that the authors simply extracted a limited amount of information relevant to the points of this manuscript from genome-wide ChIP-seq data sets without providing a general description or any general analyses of those ChIP-seq data sets. The authors response was to provide a limited amount of data to address some of the specific questions listed in the third critique of the original review and to decline to provide any additional general information about the datasets because a different manuscript that will provide this information is in preparation.

We understand this reviewer's concerns and we hope this revised version will be satisfactory.

Original Critique #1:

At issue are the examples of changes in short- and long-range interactions due to Ldb1 knockdown shown in Figure 5g. The authors concluded that “visual inspection of the HiChIP matrices confirmed the reduced long-range interactions at loci characterized by Pax3-dependent Ldb1 recruitment (Dbx1 and Cdon loci in Fig. 5g)”, but the data is not convincing and seems to indicate a qualitative change in the location of interactions, not a loss of interactions. A clear decrease in the number of interactions is not apparent despite the table indicating that ~190 loops are lost with Ldb1 knockdown. Is there a better way to convey these findings?

We have reanalyzed our data using a novel algorithm (FitHiChIP) optimized for HiChIP and PLAC-seq datasets (<https://ay-lab.github.io/FitHiChIP/>). As shown before, we detected a reduced number of long-range interactions upon Ldb1 knockdown (9,565 and 6,885 loops in control and Ldb1-knockdown cells, respectively - Revised Figure 5f). To better convey these findings, as suggested by this reviewer, we have included separate contact matrices for the control (shSCR) and Ldb1-knockdown (shLdb1) datasets in Revised Figure 5g and Supplementary Figure 6h. These revised panels also include the looping interactions detected at these loci and the change in intensity observed between shSCR and shLdb1 normalized matrices.

Original critique #2:

The point of this critique was that the authors had not directly connected looping between Pax3 bound sequences and expression of those genes. The response to the critique was description of an experiment where the DNA binding domain of Pax3, which does not have any additional functions, was fused to Ldb1. The fusion protein recapitulates the myogenic properties of wildtype Pax3. While this experiment, as the authors state in the rebuttal letter, “shows that recruiting LDB1 is sufficient for the activity of Pax3”, it does not demonstrate that looping mediates transcription. It demonstrates that targeting Ldb1 to Pax3 bound loci mediates the function of Pax3. The authors could have attempted to determine the requirement for specific Pax3 sites in loop formation and in mediating gene expression at any myogenic locus by manipulating endogenous sequences or even by re-creating a loop using myogenic sequences in a reporter assay of some sort. Alternatively, the Hi-ChIP data from the control and Ldb1 knockdown cells could be mined to identify loops at myogenic genes. If the loops were altered and/or less frequent in the knockdown cells, this would at least extend the correlation between looping and myogenic gene expression, and the authors could modify their conclusions to indicate that a correlation exists.

To address this point, we performed HiChIP in non-induced and 1-day induced Δ TAD-Ldb1 cells. As previously shown, Δ TAD-Ldb1 recapitulates Pax3 function by inducing several target genes (Figure 6C) upon 1-day induction. As shown in Revised Figure 6f, our HiChIP studies using an anti-Ldb1 antibody to pull-down Δ TAD-LDB1 show that gene expression changes are accompanied by establishment of looping interactions, similar to the ones observed upon 6-day Pax3 induction (please compare Figure 6f to Figure 1e and Supplementary Figure 2a). Therefore, we conclude that Δ TAD-Ldb1 forces the formation of loops and induces transcription. Of note, a study (or “Deng and colleagues”) has documented a similar approach using a Zinc-finger DNA binding domain fused to Ldb1 to rescue globin gene expression in Gata1-null erythroblasts through

forced chromatin looping (<https://www.ncbi.nlm.nih.gov/pubmed/22682246>). Because Pax3 is not endogenously expressed during our differentiation protocol (<https://www.ncbi.nlm.nih.gov/pubmed/23081715>), the observed changes in gene expression are solely dependent on Δ TAD-Ldb1 induction.

Original critique #3 (also relevant to critique #8):

In response to the request for results and analysis of the Pax3 ChIP-seq data set, the authors provided the number of peaks on days 1 and 6 and the % of peaks in different genomic locations. The data provide assurance that the conclusions are not based on small number of peaks, but general features of the data set, both technical and biological, remain unknown. For example, a major conclusion of the manuscript is that Pax3 targeting of Ldb1 promotes myogenesis, but the authors do not provide information about the locations of Pax3 binding sites relative to myogenic genes across the genome. So how can the reviewer evaluate the conclusion?

Similar critiques were made about the Ldb1 and Smc1 ChIP-seq datasets. The authors' response was to provide Fig. S6a, which features a Venn diagram comparing the number of peaks and numbers of overlapping peaks at day 1 in induced and uninduced cells. This provides no specific information about the differences between the induced and uninduced state or any information about the locations/identities of these peaks. It is a collection of numbers that is of limited biological meaning.

A manuscript "in preparation" could be a year or two from publication. This reviewer finds the concept of generating and using a genome-wide data set but reserving key features and results for an undetermined amount of time to be unacceptable. If there is not a journal policy that all data needs to be presented, there should be.

As mentioned in the beginning of this letter and in response to point #2 of Reviewer 1, the manuscript describing Pax3 ChIP-seq data has been accepted for publication in PLOS Biology. Raw and processed genomic data for this study are now available in the GEO database under the accession number GSE126362. Pax3 ChIP-seq data are available under the accession number GSE125203. The lists of looping interactions for all the HiChIP datasets detected by FitHiChIP have been included in Supplementary Table 1. Peak annotation data are provided in Supplementary Table 3.

Minor point:

Line 102 of the text refers to Fig. S1D. Did the authors mean to refer to Fig. S2B?

Text has been revised accordingly.

Reviewer #3:

The authors have adequately dealt with this reviewer's queries.

We thank this reviewer for the constructive remarks.

Reviewer #4:

I am satisfied with the revision and have no further questions.

We thank this reviewer for the constructive remarks.

Reviewers' Comments:

Reviewer #1:

Remarks to the Author:

The authors have satisfactorily answered the queries raised and have significantly improved the manuscript that I consider suitable for publication.

Reviewer #2:

Remarks to the Author:

This 2nd round of revisions from Magli et al. has addressed the major concerns raised previously by Reviewer 2.

#1

For figure 5g, thank you for re-analyzing the HiChIP data and highlighting the regions that show DNA interaction changes with Ldb1 knockdown. The Ldb1 knockdown effect appears to be modest overall, however the changes observed at the highlighted loci are apparent, if modest, and thus the data is sufficient to support the authors' claim that there are reduced long-range interactions at Pax3-dependent loci with Ldb1 knockdown.

#2

The new HiChIP data included for figure 6 is impressive, especially considering that these cells are only induced for 1-day and resemble the 6-day induced cultures from Figure 1. The new data provide convincing evidence that delta-TAD(Pax3)-fused with Ldb1 can induce DNA looping events that correlate with increased transcription and restore myogenesis.

#3

The authors' recently published paper resolves the prior concerns regarding lack of information about the Pax3 ChIP-Seq analyses, and the authors provided a GEO number for the other ChIP-seq datasets that are unique to this manuscript.